# Host biology, ecology and the environment influence microbial biomass and diversity in 101 marine fish species

Jeremiah J. Minich[1] ✉, Andreas Härer [2], Joseph Vechinski[3], Benjamin W. Frable[4], Zachary R. Skelton [5], Emily Kunselman [3], Michael A. Shane[6], Daniela S. Perry[7,8], Antonio Gonzalez[7], Daniel McDonald [7], Rob Knight [7,8,9,10], Todd P. Michael [1] & Eric E. Allen[3,5,9,11]

Fish are the most diverse and widely distributed vertebrates, yet little is known about the microbial ecology of fishes nor the biological and environmental factors that influence fish microbiota. To identify factors that explain microbial diversity patterns in a geographical subset of marine fish, we analyzed the microbiota (gill tissue, skin mucus, midgut digesta and hindgut digesta) from 101 species of Southern California marine fishes, spanning 22 orders, 55 families and 83 genera, representing ~25% of local marine fish diversity. We compare alpha, beta and gamma diversity while establishing a method to estimate microbial biomass associated with these host surfaces. We show that body site is the strongest driver of microbial diversity while microbial biomass and diversity is lowest in the gill of larger, pelagic fishes. Patterns of phylosymbiosis are observed across the gill, skin and hindgut. In a quantitative synthesis of vertebrate hindguts (569 species), we also show that mammals have the highest gamma diversity when controlling for host species number while fishes have the highest percent of unique microbial taxa. The composite dataset will be useful to vertebrate microbiota researchers and fish biologists interested in microbial ecology, with applications in aquaculture and fisheries management.

The Earth may contain around $10^{30}$ microbial cells[1] distributed across $10^{12}$ species of microbes (Bacteria and Archaea)[2], albeit many of these species likely share functional metabolic redundancy[3,4]. Several efforts have sought to describe these communities as they relate to their environmental biomes[5,6], however few studies have focused on non-

human vertebrates[7–9]. The host-associated gut microbiota in vertebrates is shaped by a variety of biological factors including host phylogeny, diet, and age, along with environmental or ecological factors such as geography, habitat, and climate, whereas less is known about other body sites[7,8,10–13]. Of the large meta-analyses that have sought to

[1]The Molecular and Cellular Biology Laboratory, The Salk Institute for Biological Studies, La Jolla, CA 92037, USA. [2]School of Biological Sciences, Department of Ecology, Behavior, & Evolution, University of California San Diego, La Jolla, CA 92093, USA. [3]Center for Marine Biotechnology and Biomedicine, Scripps Institution of Oceanography, University of California San Diego, 9500 Gilman Drive, La Jolla, CA 92093-0244, USA. [4]Marine Vertebrate Collection, Scripps Institution of Oceanography, University of California San Diego, 9500 Gilman Drive, La Jolla, CA 92093-0244, USA. [5]Marine Biology Research Division, Scripps Institution of Oceanography, University of California San Diego, La Jolla, CA 92093, USA. [6]Hubbs-SeaWorld Research Institute, 2595 Ingraham Street, San Diego, CA 92109, USA. [7]Department of Pediatrics, University of California, San Diego, La Jolla, CA 92093, USA. [8]Department of Bioengineering, University of California, San Diego, La Jolla, CA 92093, USA. [9]Center for Microbiome Innovation, University of San Diego, California, La Jolla, CA 92093, USA. [10]Department of Computer Science, University of California, San Diego, La Jolla, CA 92093, USA. [11]Department of Molecular Biology, School of Biological Sciences, University of California San Diego, La Jolla, CA 92093, USA. ✉e-mail: jeremiah.minich@gmail.com

evaluate vertebrate host microbial diversity, most focus exclusively on hindgut or stool from terrestrial animals from captive (zoo) environments whereas studies from wild animals may yield different findings[7,12,14]. Fishes, despite being the most phylogenetically diverse vertebrates, are severely underrepresented in these studies[7,12]. This underrepresentation is a critical concern because many hypotheses and patterns have arisen from these studies, including phylosymbiosis and contributions of diet driving community assemblies, yet body sites outside the gut are largely ignored and aquatic animals are insufficiently sampled to establish a generality of conclusions. Phylosymbiosis, the association of microbial composition with the host phylogeny, in vertebrates has been shown to occur in the hindgut[15] and the skin[16] of mammals, but no studies have tested this hypothesis in fishes across multiple body sites. When considering habitat differences across vertebrates, fishes are exposed to ~6–9 orders of magnitude more microbes by virtue of their aquatic existence ($10^5$ cells/ml)[17] as compared to terrestrial vertebrates breathing air ($10^2$–$10^6$ cells/m$^3$)[18]. Thus, since fishes are exposed to much higher concentrations of microbes throughout their even longer evolutionary history, it is likely their level of phylosymbiosis will differ from terrestrial vertebrates.

Fishes include several broad classes collectively representing the largest diversity of vertebrates (>35,000 species): Agnatha (jawless), Chondrichthyes (cartilaginous), Sarcopterygii (lobe-finned), and Osteichthyes (bony). Fish differ from mammals in that they generally breathe and excrete nitrogenous waste from the gills. Other body sites, including the skin, have evolved unique immune functions such as enhanced mucosal production for pathogen defense[19]. Body site is frequently one of the strongest predictors when comparing single fish species including Rainbow Trout[20], Atlantic Salmon[21], Pacific Chub Mackerel[10,21], Yellowtail Kingfish[22], and Southern Bluefin Tuna[23]. A study which analyzed microbiota from 13 species of fish from 5 families all caught in a bay, also found the body site was a major driver of microbial composition[24]. This study however combined the entire gut microbiota from midgut and hindgut.

Environmental factors such as seasonality, salinity, and geography along with diet can influence the microbiota compositions in fish[10,11,25]. Gut microbiota of vertebrates including fishes are often differentiated by trophic level for beta diversity[11,26–28]. Gut lengths are longer in herbivorous fishes which is hypothesized to increase digestion efficiency but also differ across habitats[29,30]. Hindgut fermentation in the guts of herbivorous fishes influences the microbial community composition[31]. In soil-associated invertebrates, higher trophic level was linked with higher microbial diversity and novel microbial species[32]. In humans however, increased fecal microbial diversity is associated with higher consumption of plant diversity[9]. One outstanding question is if microbial diversity is higher in fishes from lower trophic levels? A less studied environmental factor which might influence fish microbiota is the habitat. Habitats can be defined by the benthic depth, the benthic substrate, the salinity, or general ecosystem structure (e.g., bay, intertidal, reef, pelagic).

This study aimed to answer three main questions to understand the ecological and biological drivers of the microbiota of fishes: (1) what are the primary factors that influence host-associated microbial communities including diversity and microbial biomass for marine fishes with a focus on anatomy (body site location), physiology (e.g., trophic level, swim performance), environment (e.g., benthic zone, habitat substrate, ecosystem structure, climate zone, etc.), and host phylogeny; (2) where do these microbes originate (e.g., sea water, sediment, host, or unknown); and (3) is microbial diversity greater in fishes compared to other vertebrates considering fishes are more genetically diverse. We sampled and analyzed the microbiota from the four primary fish mucosal body sites (gill, skin, midgut, and hindgut) for 101 species (28 orders, 55 families, and 83 genera) of marine fishes from Southern California (SoCal; Eastern Pacific Ocean "EPO"), which represent ~25% of the local marine fish diversity, to quantify impacts of host phylogeny, trophic level, habitat type, swim performance, and

body site on microbial biodiversity and biomass. We also included gill samples from 17 species of fishes from the Atlantic, including two species also in the EPO dataset, bringing the total to 30 orders, 61 families, 96 genera, and 116 species.

Here, we show that anatomy (body sites) is the primary driver of host-associated microbiota with the midgut having the highest overall diversity. Microbial biomass in the gill is negatively associated with larger pelagic fishes (caught offshore) suggesting a potential physiological adaptation or trait in the host to be further explored. We describe patterns of phylosymbiosis occurring in multiple body sites including the hindgut, gill, and skin microbiota communities of fishes. In our comparison to multiple vertebrate classes, we show that fishes have the most unique set of microbes (92% not found in amphibians, reptiles, birds, or mammals) but that hindgut microbial diversity is highest in mammals which emphasizes the strong niche differentiation potentially from hindgut fermentation. Hindguts have the lowest cumulative gamma diversity in fishes; thus, our study highlights the importance of expanding microbiota studies to body sites other than the hindgut or feces.

## Results

### Sampling and microbial biomass estimation

From March 2018 through September of 2020 fish were collected during both directed sampling efforts along with passive sampling, namely through donations from recreational anglers. Fish were caught from a diverse range of nearshore and offshore habitats along with a range of depths (0 to 500 m) in the EPO and Western Atlantic (Fig. 1a, b). Standard biometric measures were taken for all fish. For the 101 species from the EPO, a total of four body sites (gill, skin, midgut, and hindgut) were sampled for microbiota analysis whereas only gill samples were processed for the Atlantic species subset (15 additional unique species) (Fig. 1c). A final table with all of the fish used in the study alphabetically sorted by order, family, and then species name along with corresponding pictures of the fish and a gill sample can be found in Supplementary Fig 1.

### Microbial biomass estimation

KatharoSeq was applied to determine the limit of detection of the assay, whereby the total post-deblur read counts of the positive DNA extraction controls were compared to the relative abundance of the known targets (Fig. 2)[33]. The 0.9 threshold was applied and the read number at $y = 0.9$ was determined to be 1150 reads (Fig. 2a). Thus, any samples with less than 1150 reads were excluded from the fish microbiota project (FMP) analysis. Since a subset of the standards had known cell concentrations, we then determined the limit of detection of the assay based on cell counts (in addition to the read counts). At $y = 0.9$, the number of input cells to the DNA extraction was estimated to be 15.95 (Fig. 2b). Since the positive controls (Bacillus subtilis and Paracoccus spp.) used in extraction generally have a high 16S copy number (~10 rRNA copies per genome), we estimated the limit of detection of the assay to be between ~16–160 microbial cells. Next, we log transformed the reads and known cell quantities to generate a model that enables one to predict cell counts from read counts ($p < 0.0001$, $R^2 = 0.8668$, slope of line = 3.497) (Fig. 2c). We used this equation to estimate the microbial biomass for each sample in the FMP dataset and then extrapolated for the total volume (ul) in the extraction and finally normalized by the mass (g) of tissue used in the extraction to get a final value of microbial cells per g of tissue. The actual distribution of taxonomy (target controls shown) within the positive titrations is displayed (Fig. 2d). Using the 1150 read cutoff, we excluded any samples in the fish microbiome project (FMP) dataset with less than 1150 reads which overall yielded a high success rate across the various sample types (Fig. 2e).

Since using a non-rarified (or raw count) dataset is somewhat contentious in the field due to the argument of not knowing absolute

abundances, and although we showed that estimating actual microbial abundances is feasible, we performed additional beta diversity testing to determine how these two strategies may impact interpretation of results (Supplementary Fig. 2). Specifically, we either rarified all samples at 1150 reads (excluding samples with less than 1150 reads) or we simply excluded samples with less than 1150 reads, keeping the raw counts (non-rarified). We also tested the effects of removing chloroplast reads from the dataset, which is a common contaminant in aquatic microbiota datasets. When comparing overall trends in the dataset, the order of significant drivers of the microbiota was generally conserved and not altered when comparing both processing methods. On a per factor basis, for Unweighted UniFrac specifically, certain factors had a slightly higher F-stat for the rarified version vs. the non-rarified, but the differences were minimal (Supplementary Fig. 2a). The decision to remove or retain chloroplasts did not change the effect size or F-stat for Unweighted UniFrac. For Weighted UniFrac the decision to rarefy or not was even less drastic with all orders of important factors remaining unchanged. Moreover, removing chloroplasts generally did not influence the order except for a flip between habitat_depth_level1 and substrata_group, whereby the chloroplasts would have a stronger

influence of community differentiation when comparing shallow (neritic), midwater mesopelagic, and bathypelagic zones. Therefore, we proceeded by using the non-rarified dataset with samples having less than 1150 reads and removed chloroplasts. The final FMP dataset includes a total of 373 successful samples including 107 gill samples, 89 skin, 94 midgut, and 85 hindgut samples (Fig. 2e). The details of the actual beta diversity statistical significance of the various fish related metadata is discussed later.

## Factors influencing alpha diversity and biomass of the fish microbiota

We evaluated the primary factors that influence the microbial communities in the marine fish mucosal samples. First, we compared the alpha diversity metrics (Chao1, Faith's Phylogenetic Diversity, and Shannon) and microbial biomass (estimated microbial cells per g tissue) across sample type (body site) from all fishes to determine if certain body sites had unique microbial signatures. For all three alpha diversity metrics: Chao1 (Fig. 3a, $p < 0.0001$ KW = 56.58), Faith's PD (Fig. 3b, $p < 0.0001$, KW = 45.95), and Shannon (Fig. 3c, $p < 0.0001$, KW = 47.43), there were significant differences across body sites. For

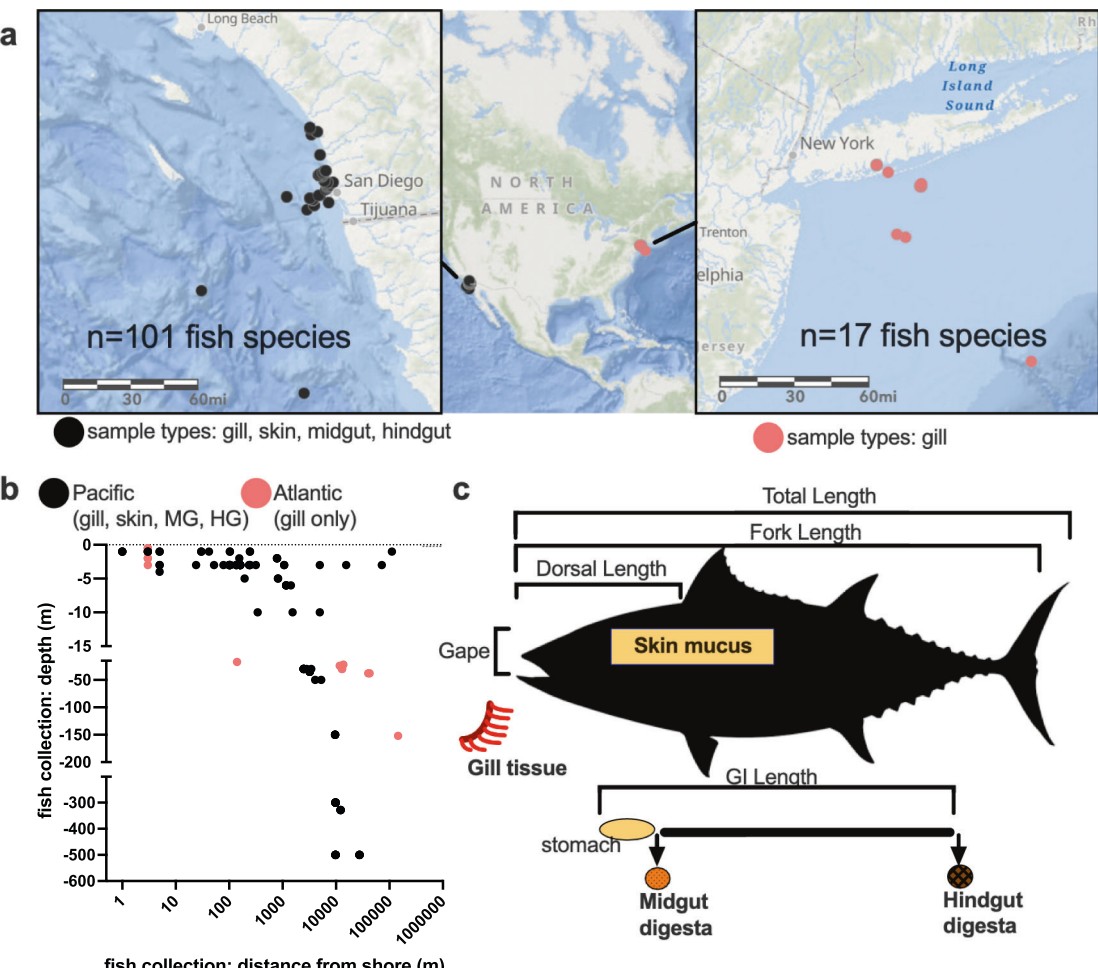

**Fig. 1 | Sampling design of 116 species of marine fish. a** Using ArcGIS to depict the general area from which fish were sampled: black dots indicate the locations of the 101 unique species of marine fish sampled from the California Current Ecosystem in the Eastern Pacific Ocean primarily in the waters of San Diego CA. Red circles depict the locations of an additional 17 species of fish (15 unique species with 2 species duplicates) collected from the Western Atlantic primarily in the waters of New York. When multiple species of fish were caught in the same location, a single circle is used to indicate the location. **b** Fish were sampled across a gradient of depth and distances from shore. **c** Biometric measurements

taken for nearly all fish include total length, fork length, mass, gape, and GI length. Various ratios from these lengths were also calculated. Microbiota samples from the gill were primarily whole tissue specimens from the entire left second gill arch or a section of the top middle and bottom of the entire filament. Skin mucus samples were taken by scraping using a razor blade. Midgut digesta material was collected from directly posterior of the stomach or if stomach was absent, the beginning of the GI tract. Hindgut digesta samples were taken from near the anus. Image from phylopic. MG midgut, HG hindgut, GI gastrointestinal tract, m meters.

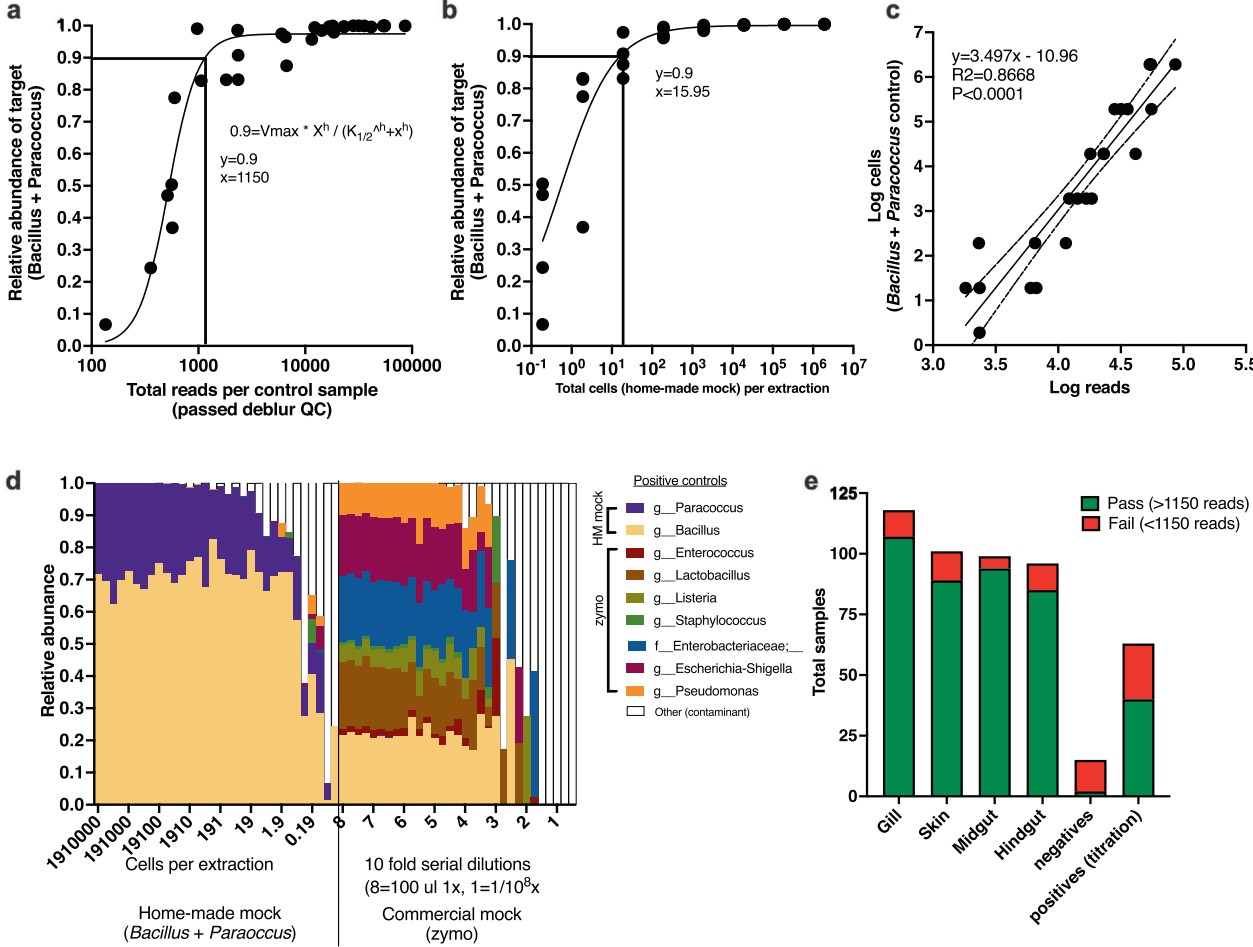

**Fig. 2 | Limit of detection, sample exclusion, and microbial biomass estimation for FMP101 dataset. a** Application of KatharoSeq formula to calculate limit of detection of microbiota plates using the Bacillus/Paracoccus mock community (1150 reads at 90%). **b** Limit of detection based on cell counts of Bacillus/Paracoccus mock community (~16 cells into extraction at 90%). **c** Model fit of the log(sequencing read counts) of positive extraction controls vs. the log cell counts of those positive extraction controls (empirically determined using plate counts). The linear regression of the line is indicative of the quality of method to estimate microbial biomass from sequencing read counts. Confidence intervals of 95% are displayed as dotted lines. This method is similar to a qPCR curve where the log (Ct) would be equivalent to the log(read counts). This equation is then used to estimate the number of "microbial density" of the existing samples which is then further normalized by the volume of the DNA extraction, biomass of material going into the extraction and finally normalized to at estimated microbial cells per gram of tissue. **d** Community analysis comparison and validation of compositionality of controls of two sets of mock community controls (section 1 = Bacillus/Paracoccus mock community; section 2 = zymo mock community). Putative contaminant g_Geobacillus identified (present in 93% of negatives and higher relative abundance as compared to positives and samples). **e** Number of samples successful across the four body sites collected from the broad fish microbiota dataset. QC quality control, g__ refers to a genus of bacteria, HM mock homemade mock or human made mixture of bacteria to use as a control whereas zymo mock = mock microbial community created by a company "Zymo".

all alpha diversity metrics, the midgut samples had the highest diversity compared to other body sites (gill, skin, hindgut), while skin had higher diversity than gill. For Shannon diversity only, skin was higher than hindgut (Fig. 3c). When comparing microbial biomass, there were no significant differences across body sites (Fig. 3d), although the range of biomass was substantial (over 6 orders of magnitude).

Various life history metrics were categorized for all fishes (Fig. 3 and Supplementary Table 1 FMP.alpha, Supp_FMP_data.dictionary). For categorical variables, a Kruskal-Wallis test was used to compare the three alpha diversity metrics and biomass (log microbial cells per gram of fish tissue). For gill samples, Shannon diversity was influenced by habitat depth level 1 and collection substrate, whereas biomass was influenced by substrata group and swim mode (Fig. 3e). Variation in skin samples was not explained by any of the categorical metadata. The midgut had the highest differences in alpha diversity explained by the metadata categories. For habitat level 1, habitat level 2, collection substrate and substrata group differences were observed for Chao1, Faith's PD, and Shannon diversity. In addition, the climate category that refers to the approximate latitudinal range and water temperature regime the fishes reside (temperate, subtropical, or tropical), biomass differed (Fig. 3e). For the hindgut samples, habitat level 1 and habitat level 2 influenced the Chao1 and Faith's PD. Shannon diversity was influenced by collection substrate and substrata group. Lastly, biomass was associated with climate, swim performance and swim mode (Fig. 3e).

Next, we assessed the continuous or numeric values using Spearman correlation across the alpha diversity and biomass metrics for each body site. Some of these metrics are ratios, for example: RIL (relative intestinal length) = total gut length/total fish length. For the trophic associated metadata categories, only the hindgut Shannon diversity was significant, and was negatively associated with high trophic level (Fig. 3f). For the swim associated metrics, acceleration was not associated with any metric across the body sites, whereas swim endurance was positively associated with microbial biomass in

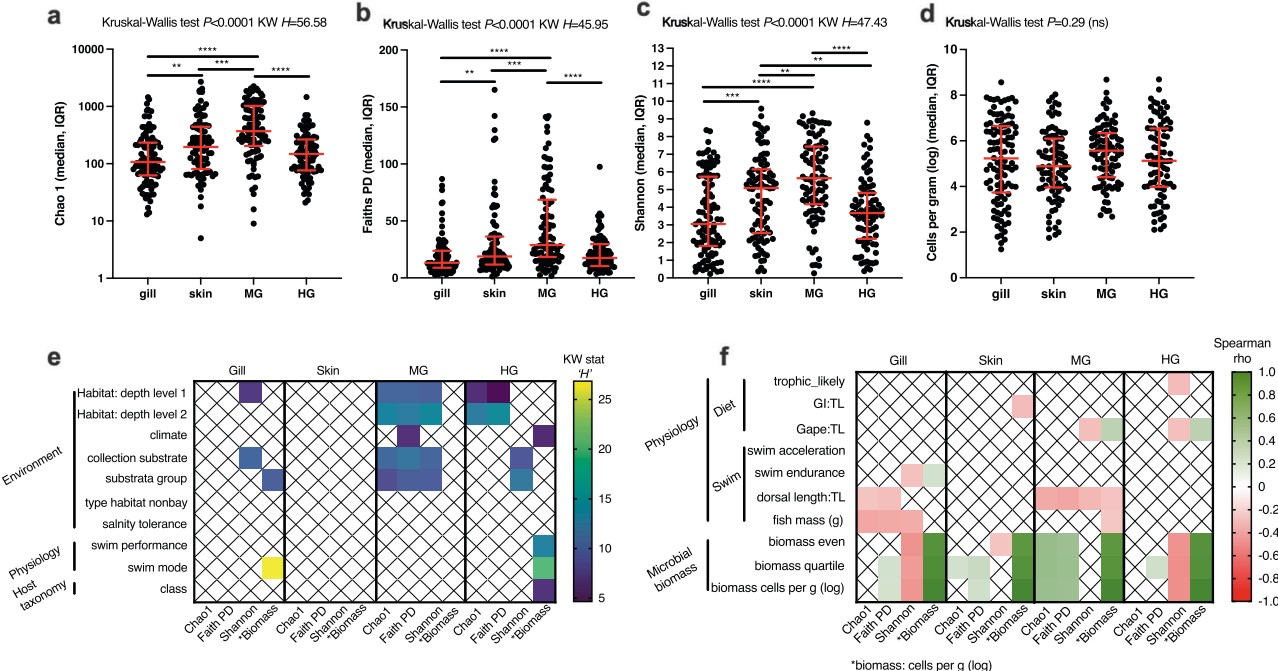

**Fig. 3 | Alpha diversity and biomass comparisons across ecological and biological gradients in marine fish.** Comparison of microbial diversity **a** "Chao1", **b** "Faith's Phylogenetic Diversity", **c** "Shannon", or **d** microbial biomass across body site (gill, skin, midgut, and hindgut). Distributions in "red" are median with interquartile range. Statistical differences determined using non-parametric testing Kruskal–Wallis test with 0.05 FDR Benjamini–Hochberg. Further testing computed for each unique body site for a variety of biological and ecological metadata categories. Metadata which is **e** categorical is tested using Kruskal–Wallis **f** whereas numeric metadata tested using Spearman correlation. Only significant associations are represented in **e** (Kruskal–Wallis $p < 0.05$) and **f** (Spearman $p < 0.05$). KW or KW stat "H" test statistic from Kruskal–Wallis test, MG midgut, HG hindgut, Faith PD Faith's Phylogenetic Diversity metric, GI:TL gastrointestinal length to fish total length "ratio", TL total length of fish.

the gill and negatively associated with Shannon diversity in the gill (Fig. 3f). The dorsal length to total length ratio was negatively associated with Chao1 and Faith's PD in the gill and negatively associated with all metrics in the midgut. This would suggest that faster swimming fishes have lower gill microbiota diversity and lower midgut microbiota diversity. For the biomass measurements, Shannon diversity in the gill and hindgut were negatively associated with microbial biomass. Faith's PD was positively associated with microbial biomass in the midgut and partially associated in gill, skin, and hindgut. Chao 1 was positively associated with microbial biomass in midgut and partially in skin and hindgut (Fig. 3f). Lastly, when comparing the total mass of the fishes against microbial biomass, there were no significant associations in the skin, midgut, or hindgut, whereas all of the gill microbiota alpha diversity metrics were negatively associated with mass of fishes (Fig. 3f).

A subset of fish samples from the neritic zone (0–200 m) and collected exclusively from the ocean (excluding samples collected from bays or estuaries) was analyzed to further evaluate microbiota associations on the fish gill. We did this to control for and reduce effects from environmental noise associated with bays such as salinity gradients and tidal flow along with temperature for deep-sea fish. Specifically we tested whether the mass of the fish or the distance from shore from which the fish was caught predicted microbiota characteristics in the fish gill. Since both fish mass and distance from shore were positively correlated (Spearman $p < 0.001$, rho = 0.43), interpretation of results should be with caution as we were not able to tease apart these confounding variables (Fig. 4a). The gill microbial biomass differed across habitats from which the fish were caught (Kruskal–Wallis $p = 0.0144$, KW = 16.31) with pelagic fish having lower microbial biomass in the gill compared to fish collected from the intertidal and subtidal zones ($p < 0.05$) (Fig. 4b). This result led us to test if either fish mass or the distance from shore had an impact on

microbial biomass or diversity as intertidal and subtidal fish are close to shore whereas pelagic primarily live offshore. Fish mass (Fig. 4c) and distance from shore (Fig. 4d) were both negatively associated with gill microbial biomass. In addition, fish mass and distance from shore were negatively associated with Chao1 (Fig. 4e, f) and Faith's PD (Fig. 4f, g). For oceanic fish living in the neritic zone, we observed that offshore fishes such as pelagics along with larger fishes have lower microbial biomass density and diversity in the gills as compared to small fish living closer to shore such as fish from the intertidal and subtidal environments.

### Factors influencing beta diversity in the fish microbiota

We generally assessed the same biological and life history traits of the fish species for microbial beta diversity (Fig. 5 and Supplementary Table 2 FMP.beta). First, we compared all samples together ($n = 373$) for Unweighted and Weighted normalized UniFrac (Fig. 5a). Sample type (gill, skin, midgut, hindgut) for both metrics were the primary predictor. Microbial biomass, habitat depth (shallow "neritic", mid "mesopelagic", and deep "greater than 1000 m"), and substrata group (refers to benthic substrate type such as soft bottom, rocky reef, deep water, pelagic, etc.) were also large predictors. Since the sample type was the biggest driver, we subsampled each body site and analyzed again. Gill samples were associated with water column depth (habitat level), benthic substrate material, and microbial biomass (Fig. 5a) for Unweighted UniFrac. There were generally fewer associations across body sites for Weighted UniFrac, but benthic substrate type was also associated with gill and hindgut communities (Fig. 5a). Trophic level did not predict any beta diversity metrics for individual body sites. However, lower trophic level fishes had a higher similarity between midgut and hindgut, whereas carnivorous fishes tended to have higher variation between midgut and hindgut (Fig. 5b). Higher trophic fishes had more differentiation between the midgut and hindgut. Although

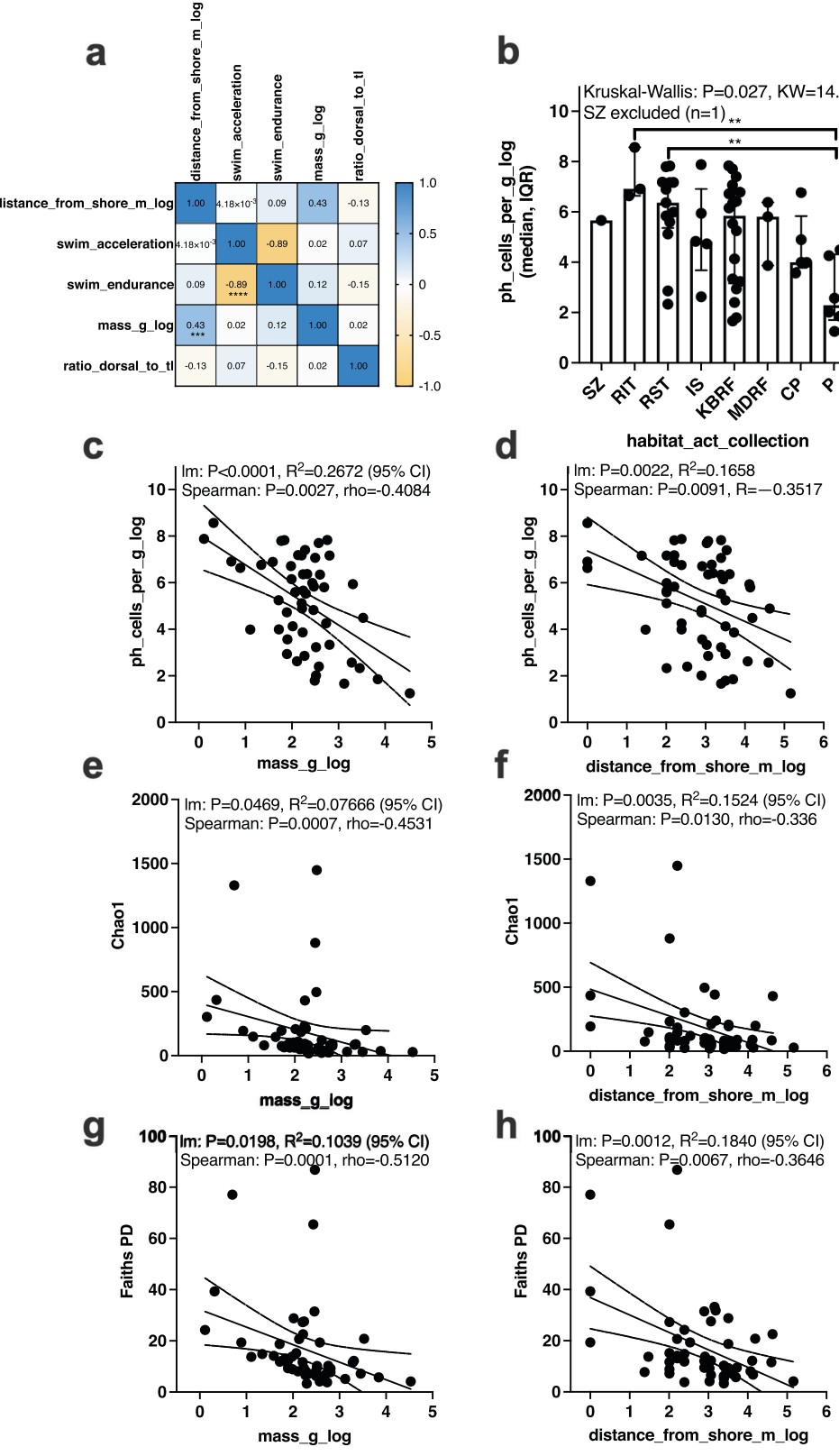

lower trophic fishes had a higher GI length to body length ratio, this suggested additional factors may be a stronger influence than gut length.

### Evidence for phylosymbiosis across multiple body sites

We evaluated if the divergence time between fish species was associated with microbiota dissimilarity using both Unweighted and Weighted normalized UniFrac across all four body sites (gill, skin, midgut, and hindgut). Using timetree.org we created a tree (Supplementary Fig. 3) to estimate phylogenetic distances between fishes. For Unweighted UniFrac distances, a presence-absence based metric, only the skin microbiota was significantly associated with fish divergence time ($p = 0.008$, $r = 0.234$, FDR = 0.05) (Fig. 6a). For Weighted UniFrac however, gill ($p = 0.011$, $r = 0.173$, FDR = 0.05) (Fig. 6b) and hindgut

**Fig. 4 | Associations between fish mass and collection location as measured by distance from shore with fish gill microbial biomass and alpha diversity.** Subset of fish from EPO and Atlantic (*n* = 54) collected from ocean (excludes bay and estuary samples) and from the neritic zone (<200 m depth). **a** Correlation matrix between sample metadata where values are rho and significance indicated by *\*p* < 0.05, *\*\*p* < 0.01, *\*\*\*p* < 0.001, *\*\*\*\*p* < 0.0001 (Spearman correlation). **b** Comparison of gill microbial biomass (log cells per gram) across habitat types from which the fish were collected. Distribution is in median and interquartile range. Statistical differences determined using non-parametric testing Kruskal–Wallis test with 0.05 FDR Benjamini–Hochberg. **c** Comparison of fish mass

and **d** distance from shore with gill microbial biomass. **e** Comparison of fish mass and **f** distance from shore with alpha diversity metrics (Chao1). **g** Comparison of fish mass and **h** distance from shore with alpha diversity metric: Faith's PD. **c**–**h** (Confidence intervals of 95% are displayed as dotted lines). habitat_act_collection refers to the metadata column name from where this habitat classification can be found…, SZ surf zone, RIT rocky intertidal, RST rocky subtidal, IS inner shelf, KBRF kelp bed rocky reef, MDRF mid depth rocky reef, CP coastal pelagic, P pelagic. Mass_g_log = log 10 (mass of the fish in grams), distance_from_shore_m_log = log 10 (distance from nearest point on shore in meters from where the fish was caught).

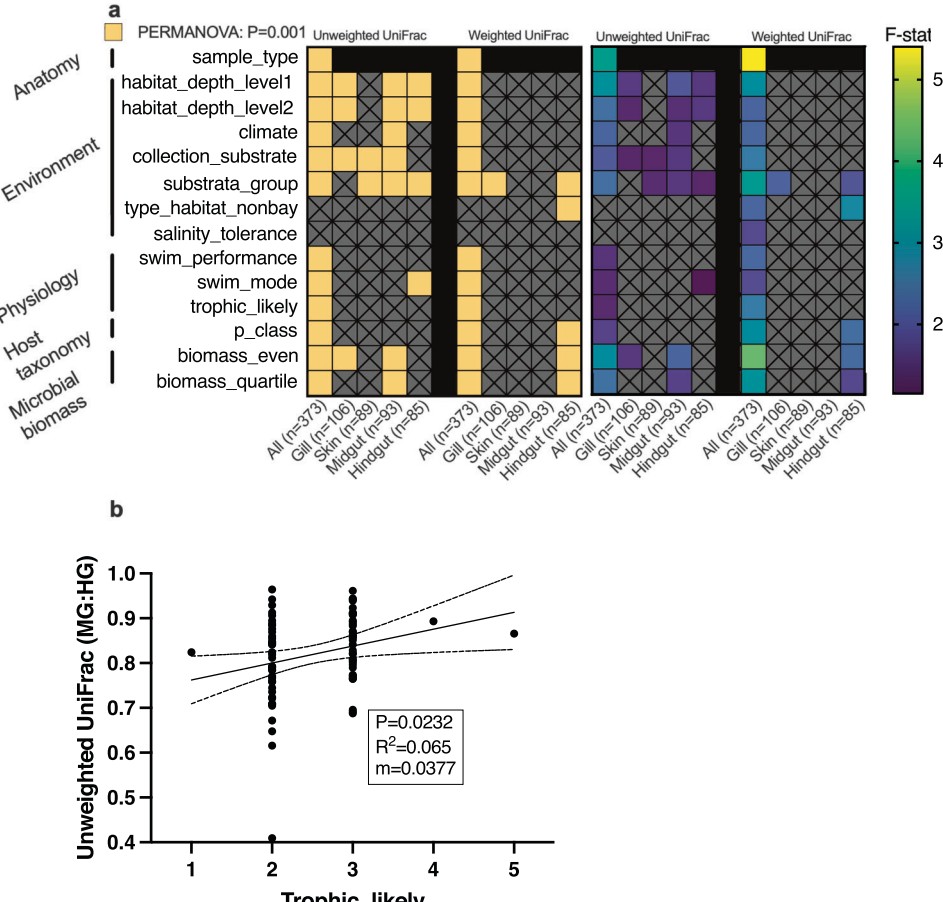

**Fig. 5 | Biological and life history drivers of mucosal microbiota in diverse sampling of marine fish from Southern California. a** Multivariate analysis of biological and life history parameters evaluated using unweighted and weighted normalized UniFrac distances. Statistical significance (PERMANOVA *p* = 0.001) indicated by yellow blocks (left) and effect size (right). All samples compared together (all) along with individual sample types (gill, skin, midgut, hindgut).

**b** Impact of trophic level on similarity between midgut and hindgut (within a species) (linear model: *p p* value, m slope, dotted lines are 95% confidence interval). F-Stat test statistic used in PERMANOVA analysis, all row names in **a** are metadata column names used in the analysis, MG midgut, HG hindgut, Gen. Weighted UniFrac generalized weighted UniFrac.

(*p* = 0.008, *r* = 0.306, FDR = 0.05) (Fig. 6c) were significant whereas skin (*p* = 0.092, *r* = 0.133, FDR = 0.05) and midgut (*p* = 0.341, *r* = 0.035) were not significant. Overall, evolutionary distance in fishes was primarily associated with weighted distances rather than unweighted, suggesting that microbes with higher relative abundances had a stronger predictive ability of host genetic similarity. In addition, within the weighted comparisons, hindgut had the strongest association followed by gill. Thus, we concluded that both host phylogeny along with the environmental signal (e.g., habitat, benthic substrate, water column depth) were factors important for shaping the mucosal microbial communities of fishes.

Since our data show that fishes that are more similar to each other (shorter phylogenetic distance or branch length) have a more similar skin, gill, and hindgut microbiota, we wanted to explore how

phylosymbiosis could be used in the discovery of probiotics. If vertebrates have co-evolved to some extent with their microbiota, one could speculate that strains from genera that contain known probiotics found in fishes would have higher performance (improved ability to adhere and colonize to mucosal environments) in fishes as compared to the application of allochthonous terrestrial-derived probiotics applied to fishes. Therefore, we next explored the extent by which taxa from genera that contain known probiotics (*Bacillus* and *Lactobacillus*), were present across fish body sites. *Bacillus* was found in a higher frequency of fish species across body sites (gill = 48.6%, skin = 48.3%, midgut = 67%, hindgut = 36.5%; percent of species with *Bacillus* present) than *Lactobacillus* (gill = 16.8%, skin = 15.7%, midgut = 19.2%, hindgut = 5.9%; percent of species with *Lactobacillus* present) (Supplementary

**a**

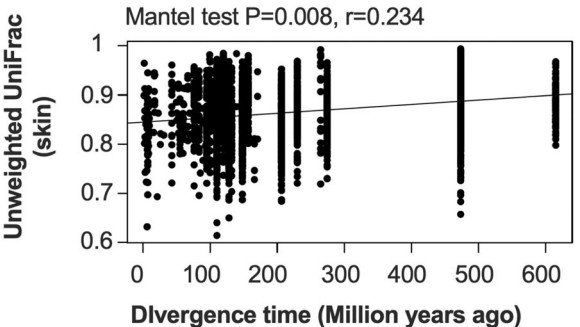

**b**

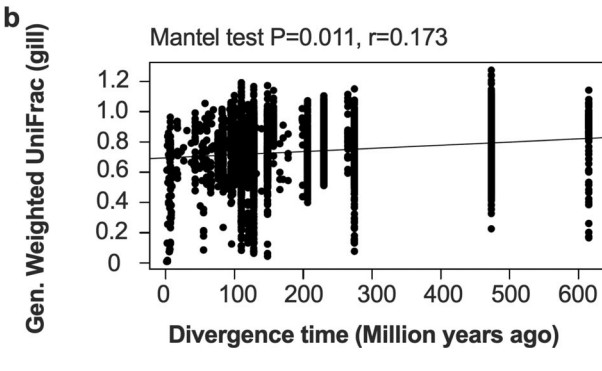

**c**

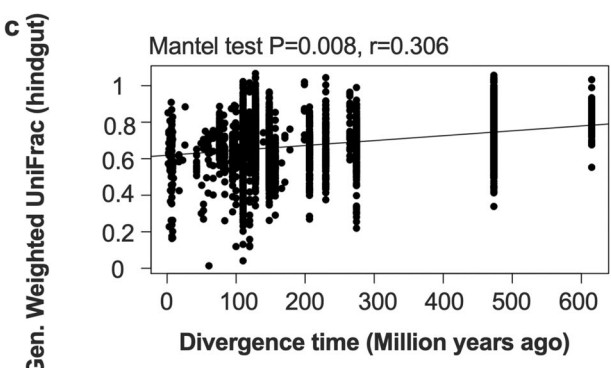

**Fig. 6 | Evidence for phylosymbiosis across fish body sites.** Effect of evolutionary distance (low divergence time indicates a short branch length or similar fish species) of all fish compared to **a** skin unweighted UniFrac distance, **b** gill generalized weighted UniFrac distance, **c** hindgut generalized weighted UniFrac distance. Comparisons performed using Mantel test with multiple testing by FDR. Divergence time between fish species calculated using timetree.org. Gen. Weighted UniFrac generalized weighted UniFrac.

Fig. 4a). Both *Bacillus* and *Lactobacillus* were found in the midgut in a higher number of fish species than other body sites. However, *Bacillus* and *Lactobacillus* made up a very small fraction of the overall community (<5%), thus future work should include enrichment methods in addition to metagenomics to describe these species (Supplementary Fig. 4b).

### Quantifying the role of sea water and marine sediment as microbial sources for fish mucus colonization

We performed an analysis using SourceTracker2 to better understand the role of the environment in shaping the microbiota of fish mucosal sites. Specifically, we included 108 marine sediment samples and 60 paired sea water samples from a 10 km transect in San Diego including samples from the beach, sandy

soft bottoms, rocky reefs, and bay sand/mud (Fig. 7a). The sediment and sea water samples were set as replicate sources with all fish samples included as unique sinks. Across all fish mucosal sites, there appeared to be a gradient of importance with beach sand having the lowest overall microbial contribution to fish mucosal sites followed by marine sediment and lastly marine sea water having the highest contribution of microbes. However, the majority of microbes were still of unknown origin (mean: gill 88.95%, skin 79.06%, midgut 77.03%, hindgut 89.93%) (Fig. 7b). We then asked if certain body sites were more likely to have known microbial sources. Fish body sites differed in the proportion of ASVs derived from sea water sources ($p < 0.0001$, KW = 39.66, Fig. 7c) with midgut generally having the highest amount of sea water microbes followed by skin, hindgut, and gill. Fish body sites also differed in the proportion of ASVs that were derived from marine sediment sources ($p < 0.0001$, KW = 23.41, Fig. 7d) again with midgut having the most sediment ASVs followed by skin, hindgut, and then gill. We next tested if within a given body site, there was a difference in the proportion of microbes originating from sea water or sediment. For all body sites, microbes originated more from sea water sources than marine sediment (Mann–Whitney $U$ test, gill $p = 0.0215$, skin $p = 0.0157$, midgut $p < 0.0001$, hindgut $p = 0.0148$) (Fig. 7e). Despite the significance, there remained a large range of values across fish species, thus we explored if certain life history traits explained when a fish had higher proportions of microbes originating from sea water or sediment. In comparing the continuous variables for each body site, we found that in the gill and midgut samples, microbial biomass was negatively associated with a SW:SED ratio (enrichment of sediment microbes as compared to sea water). The dorsal:TL ratio (indicating acceleration potential or fast swimming fish) for the gill, skin, and hindgut samples was positively associated with the SW:SED indicating fish that swam fast had more sea water sourced microbes in those body sites. In addition, the gape to TL ratio was positively associated with SW:SED in the skin and hindgut while trophic level was positively associated with SW:SED in hindgut. Mass and condition factor (length to mass measurement similar to BMI) for midgut, was positively associated with SW:SED (Fig. 7f). Since body shape morphology with context to swim performance was associated with SW:SED, we next compared if overall body shape as it relates to swimming (metadata column: swim_performance) was also associated. Specifically, we were interested to know if flatfish that have adapted to living in the sand had a higher proportion of microbes originating from sediment. When comparing skin mucus, swim mode differed ($p = 0.0158$, KW = 13.97). Specifically, cruiser/sprinter fish (including the mackerels, tunas, and jacks, etc.) had a higher SW:SED (median = 7.916) than flow refuging (flatfish and stingrays/skates) (median = −1.203) and maneuverer fish (deep bodied reef fish like the Kyphosids) (median = −2.794) (Fig. 7g). Based on this observation, "flow refuging" fish which includes flatfish indeed have a higher proportion of their skin microbes originating from the sediment whereas the fast swimming fishes (pelagics and coastal pelagics) have most of their skin microbes originating from the water column. One aspect we were not able to compare was the influence of prey items. Based on diet surveys, marine invertebrates were consumed by 107 species of fish while fish were the second most common prey item (Supplementary Fig. 5a). Within the invertebrates consumed, arthropods, mollusks, and annelids were the most consistent prey items (Supplementary Fig. 5b) all of which have many species globally (Supplementary Fig. 5c). By comparing the ratios of microbial origin environments (e.g., sea water vs. sediment), we can discover insights into host-associated microbial ecology in the ocean. Microbial origins of marine fish mucosal remain largely

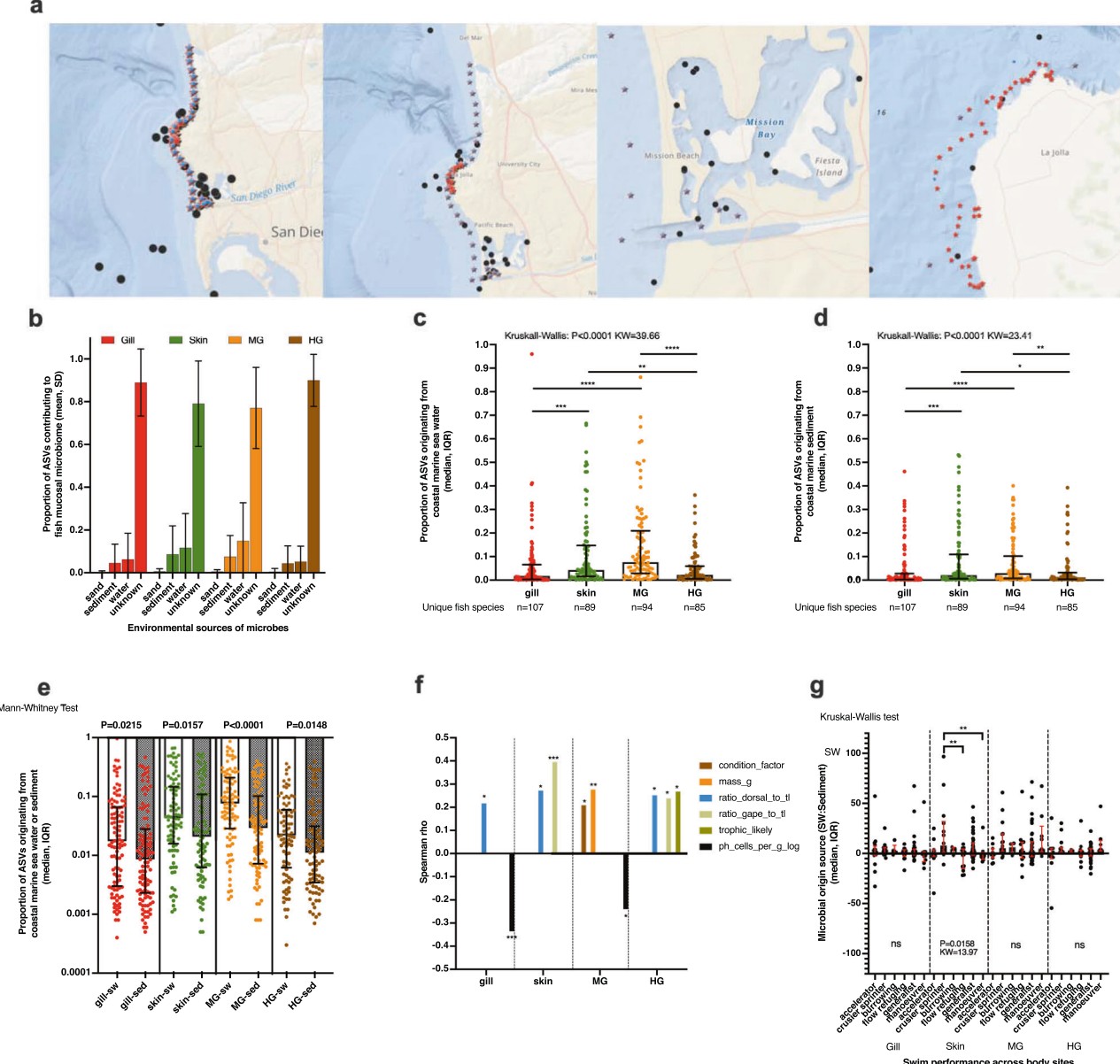

**Fig. 7 | Microbial source tracking analysis. a** Microbial sources of 60 sea water (blue circle) samples taken from 30 unique sampling stations from two time points are distributed on a 10 km transect from Torrey Pines beach to Mission Bay. Microbial sources of 108 marine sediment samples (red stars) from San Diego coastal environment includes 60 paired samples (same locations as sea water) from the same 10 km transect along with 58 samples from the various reef habitats near La Jolla. Geographic data presented using ArcGIS. **b** Sourcetracker2 analysis of likely sources for the four body sites of the fish comparing contributions of beach sand, marine sediment, sea water, and "unknown". Unknown refers to microbes from an unknown source which could include diet and other animals or locations not sampled. **c** Specific microbial contributions of sea water to the four mucosal body sites and **d** specific microbial contributions of marine sediment to the four mucosal body sites **b–d**: distribution is in median and interquartile range. Statistical differences determined using non-parametric testing Kruskal–Wallis test with 0.05 FDR Benjamini–Hochberg. **e** Proportion of microbes (distribution is in median and interquartile range) likely originating from the sea water vs. sediment for each unique body site (sea water vs. sediment pairwise comparison for each body site using Mann–Whitney test $p < 0.05$). **f** Spearman rho values from comparisons of the ratio of sea water "SW" and marine sediment "SED" against various continuous fish life history metadata variables for each unique body site (Spearman correlation $p < 0.05$). **g** Comparison of the SW:SED ratios across the habitats from which the fish live. Comparisons performed on each unique body site (Kruskal–Wallis test, $p < 0.05$). *$p < 0.05$, **$p < 0.01$, ***$p < 0.001$, ****$p < 0.0001$, ASV amplified sequence variant -unique sub-Operational Taxonomic Unit, SD standard deviation, MG midgut, HG hindgut, KW Kruskal–Wallis test statistic "H", IQR inter quartile range, SW sea water.

unknown, but ecological factors, such as habitat type, may influence the extent of microbial colonization from sea water or sediment.

## Gamma diversity analysis across vertebrates

We performed a quantitative synthesis focusing on hindgut fecal samples that were the body site of broadest interest in the field and thus had the most samples. Hindgut microbial gamma diversity was compared on rarified data from a total of 569 unique species across the five vertebrate classes (fishes $n = 73$, birds $n = 216$, mammals $n = 208$, reptiles $n = 52$, and amphibians $n = 20$) to test whether total microbial diversity "gamma" was greater in animals with an older evolutionary history (Fig. 8a). If animals have co-evolved with their microbiota, one could hypothesize that older lineages will have had more time to optimize these microbial relationships. This could include removal of detrimental taxa or enrichment of beneficial taxa. If microbiota

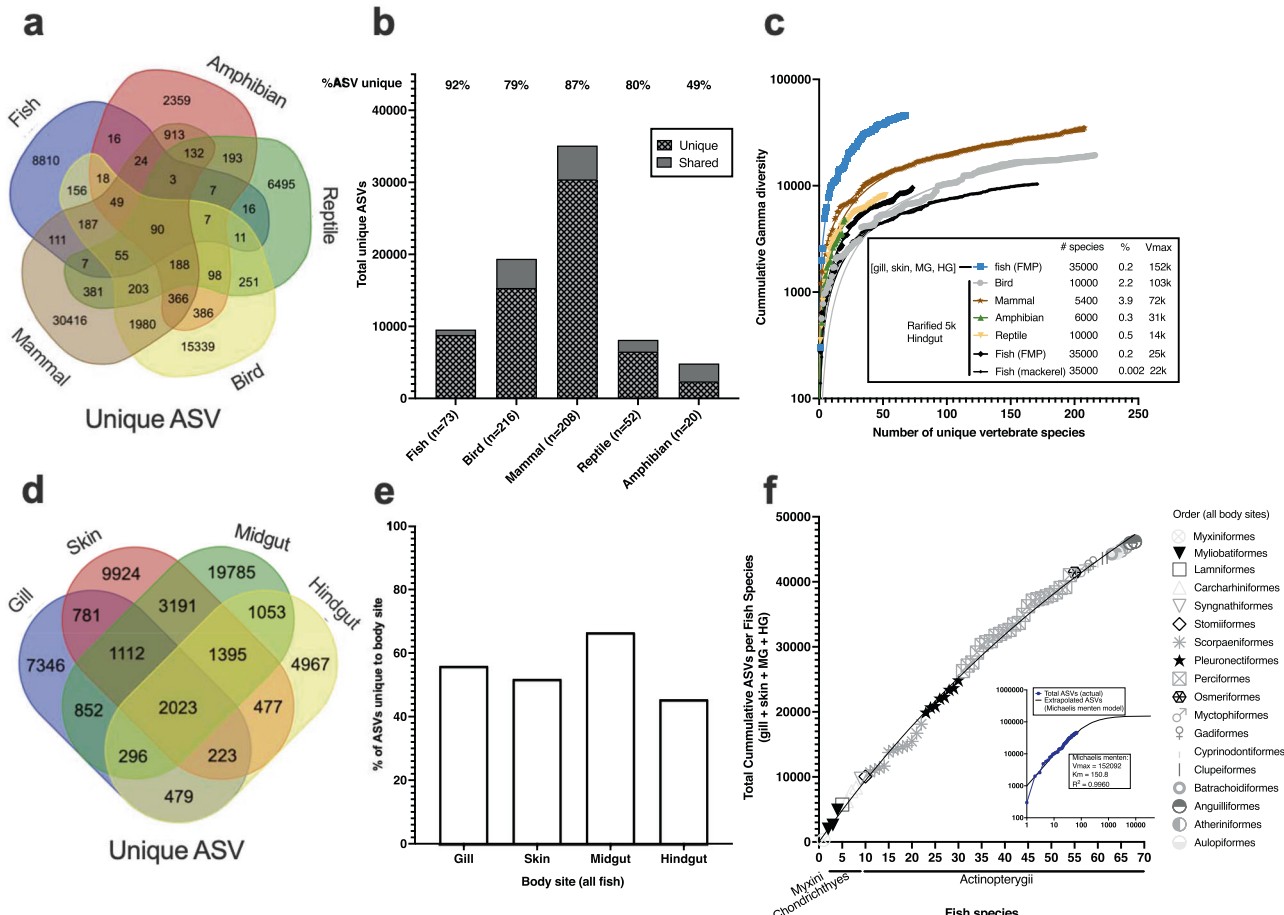

**Fig. 8 | Total microbial diversity across vertebrate hindguts and within multiple body sites of fish. a** Hindgut microbiota samples from 569 species of vertebrates were rarified to 5000 reads and unique or shared ASVs determined for each class. **b** The percentage of unique ASVs only found in a given class (not shared in other classes) as compared to the total ASVs within that class. **c** Rarefaction of cumulative gamma diversity as a function of unique vertebrate species. Included is a single fish

species, S. japonicus, sampled over 3 years "black dots" and the unrarefied FMP samples which had detectable bacteria in all four body sites (gill, skin, midgut, and hindgut). **d** Gamma diversity of 68 fish species across four body sites. **e** Percentage of unique ASVs associated with a given body site across the 68 fish species. **f** Rarefaction curve of increasing gamma diversity (inclusive of four body sites) as a function of increasing fish species. ASV amplified sequence variant, 5k 5000.

colonization is random, then one would expect diversity to simply be a function of the environmental exposure. Mammals had the highest total number of amplified sequence variants "ASVs" (35,105), followed by birds (19,384 ASVs), and then fishes (9567 ASVs) (Fig. 8b). The majority of ASVs were unique to each broad vertebrate class (e.g., mammal, reptile, amphibian, bird, or fish) and not shared with other classes. Of the classes, fishes had the highest percentage of unique ASVs that were not found in other classes (92%), while mammals were second highest at 87% (Fig. 8b).

Since the total number of species sampled differed across vertebrate classes, we next compared total cumulative microbial diversity on an additive basis for individual species for both the rarified hindgut samples for all vertebrates along with the full unrarefied FMP dataset (including other body sites: gill, skin, midgut, and hindgut) of fishes collected from SoCal USA. When comparing hindgut samples only, among 50 different animal species, mammals had the greatest gamma diversity followed by reptiles, fishes and birds the lowest. Multiple replicates (50 fish replicates across 3 years and four seasons) of a single species of fish, *Scomber japonicus*, was included as reference for comparison purposes, although this was similar in magnitude (total unique microbial hindgut diversity at 50 replicates) to birds (Fig. 8c).

We next assessed gamma diversity strictly in fish samples that had sufficient microbial sequences in all four body sites (*n* = 68 species) based on the sample exclusion criteria calculated from KatharoSeq[33]

(Fig. 2). Because we can estimate microbial biomass on a per sample basis based on read counts for the FMP dataset, we included samples as unrarefied to enable a better estimate of total diversity. Midgut samples overall had the greatest number of ASVs unique to that body site (19,785 ASVs) followed by skin (9924 ASVs), and gill (7346 ASVs) with hindgut (4967) having the lowest overall gamma diversity (Fig. 8d). We then calculated the extent by which those ASVs made up the total diversity in a given body site (unique ASVs/total ASVs). The midgut also had the greatest proportion of ASVs which were unique to that body site (66.6%) followed by gills (56%), skin (52%), and hindgut (46%) (Fig. 8e). We conclude that hindgut samples in fishes are the body site with the lowest total microbial diversity and lowest unique body site associated diversity. Thus, comparisons across vertebrates should begin to include other body sites that may harbor more microbial biomass and diversity.

Lastly, we attempted to assess the feasibility of estimating the total number of 16S rRNA gene V4 region ASVs across all of the estimated 35,000 extant species of fishes. To do this, we performed a rarefaction plot on increasing fish diversity with total cumulative gamma diversity of fishes. Gamma diversity was calculated per fish species by combining all unique ASVs across the gill, skin, midgut, and hindgut. The fish species were arranged on the *x*-axis by class (with earliest lineages first: Myxini, Chondrichthyes, Osteichthyes) and then alphabetically by order and family. Results of the model suggested that

saturation occurred after sampling around 1000 species of fish and there was an estimated total of 152,092 total unique ASVs associated with fishes (Fig. 8f). On the contrary, visual inspection suggests a very small change in the overall trajectory that suggests an accurate estimate of gamma diversity is not possible without many more species, perhaps hundreds or even thousands of additional species. While the goodness of fit for the model is quite high ($R^2 = 0.9960$), it is entirely possible that we are severely underestimating total diversity, because saturation of the curve is not easily apparent. Also, our study was limited by focusing on marine temperate fishes and few tropical fishes so it was likely underestimating the diversity, especially because it excludes tropical coral reef associated fishes. For future research on gamma diversity across other vertebrates, we recommend comparisons of multiple body sites along with the integration of microbial biomass estimates as demonstrated in this study. The integration of microbial biomass enables a more accurate assessment of total absolute microbial diversity in a sample. By using our method, one uses all the reads in the sample to estimate diversity rather than needing to rarefy which forces one to lose or disregard sequences leading to underestimates of true diversity.

## Discussion

Here, we developed a new method to estimate microbial biomass per gram of tissue using 16S amplicon sequencing and then applied it to investigate the ecological and biological drivers of bacterial diversity associated with fish mucosal sites (gill, skin, midgut, and hindgut) across 101 fish species from the EPO along with gill samples from an additional 17 species (15 unique) from the Western Atlantic. We curated a list of both categorical and continuous metadata types to describe the biology of the fish including habitat, diet, and swim performance. We tested the effects of host phylogeny on microbial diversity. Finally, we performed an extensive quantitative synthesis of gamma diversity of marine fishes as compared to other vertebrate classes and show the importance of including multiple body sites along with microbial biomass in measures of diversity.

One of the biggest challenges in microbial ecology is estimating microbial biomass also known as total microbial load or absolute abundance. This is especially challenging in host-associated environments as the existing methods to do this including microscopy, flow cytometry, and qPCR[34] are optimized to work best on free-living systems and may have lower throughput or require immediate processing. Flow cytometry methods are challenging due to excessive clumping of cells, lower throughput, and require immediate processing of samples[34] while qPCR assays (e.g., 16S) can have many false positives due to contamination and thus co-amplification of either chloroplasts or mitochondria[35]. Another approach is to use a spike-in of known cell counts of control bacteria[36] or synthetic oligos[37] into your samples which can enable biomass estimations post sequencing. The downside to this approach is you lose sequencing depth across samples and if using cells, it is necessary to ensure that those microbes are not already found in your samples. Many studies ignore estimating biomass and instead are forced to assume equal biomass across samples. This strategy causes problems when trying to compare relative abundances of microbes across samples which has led to many computational tools being developed around compositionality[38,39]. Our method is useful and unique in that it enables one to estimate the microbial biomass (namely number of 16S copies per gram of tissue) using standard 16S high throughput sequencing and without contaminating samples with a spike-in. One caveat is that many microbes have multiple 16S copies per genome, thus one may want to perform this normalization on their sample set in the future if rrn copy number is known across the resident microbial taxa. Attempting to normalize per 16S copy number per genome however is challenging and most studies actually fail to do this, thus we have avoided this additional normalization attempt in our workflow[40]. Our results directly refute

the hypothesis that sequencing depth is random and instead show that it is highly correlated to the microbial biomass. We leverage this observation and provide a framework for consistent sample processing to enable insights into microbial biomass which should be applicable across all ecosystems including animals, plants, soil, water, and the air.

The current study evaluates a range of host and environmental variables that may influence the microbial diversity associated with a broad phylogenetic sampling of marine fishes. Among all the factors tested, the "body site" was the strongest predictor of microbial community composition followed by depth, habitat benthic substrate, and fish microbial biomass. This suggests that there are conserved anatomical or physiological aspects of the body sites across fish lineages, which select for or enrich certain microbial communities. We hypothesize this could be due to body site differences in microbial exposure, immune function, mucus chemistries, morphology, and host anatomy and physiology[41]. Microbial biomass, habitat depth 1 (shallow "neritic", mid water "mesopelagic", or deep-sea "bathypelagic or abyssal"), and substrata group (pelagic, soft bottom, rock associated, deep water benthic) were also top predictors of beta diversity across the entire dataset suggesting that habitat had an influence on the microbiota of fishes. Microbiota diversity, function, and stability is known to vary across water column depth as a response to decreasing sunlight, availability of organic matter and nutrients mediated by the microbial loop, mixing, oxygen concentration, and water temperature[42,43]. Thus, deep-sea fishes, especially those which have limited vertical migration, would have different microbial exposures at depth. The influence of substrata group on fish microbiota diversity was an interesting observation. We speculate that the benthic structure likely influences the sediment microbiota and that in turn influences resuspension of microbes and remineralization of nutrients back into the water column. Benthic complexity is a key driver of local fish diversity (e.g., kelp forest vs. inner slope) thus animal assemblages themselves will differ across varying substrates[44]. Conversely, the diversity of fish or animal assemblages will influence the diversity and magnitude of organic wastes which get deposited to the benthos thereby impacting local sediment microbial diversity. Fishes, particularly flatfish, can also influence the benthic sediment by physical resuspension[45]. To date, sediment microbiology research has focused on impacts to free-living communities in the overlying water column, with less attention to impacts on the overlying animal assemblages and vice versa. We see this as a promising area of future research to better understand the microbial loop and sediment contributions to host-associated microbial biodiversity.

Each body site had specific associations with the various ecological and biological parameters. Fish skin primarily functions as a protective barrier by preventing invasion of pathogens, but in some species the skin can have additional physiological roles such as a source of gas exchange[46]. Here, the fish skin microbiota was primarily explained by the type of bottom structure of the habitat. For shallow environments, the benthic substrate (e.g., mud, sediment, rocky reef) will likely have a stronger contribution of microbes directly to the water column as compared to deeper water systems and therefore may explain how sediment types can influence external microbiota of fishes[47,48]. Differences between the midgut and hindgut emphasize the need to further describe the physiological or abiotic factors (e.g., oxygen) micro gradients within the GI tract of fishes[49]. The gastrointestinal tract can vary in morphology and function across fishes with many differing types of stomachs, lengths, and enzyme profiles[30,50,51].

Several novel observations were made in the context of fish ecology and the gill microbiota. We observed decreasing microbial diversity (Chao1 and Faiths PD) in the gills of larger fishes and fishes that were morphologically associated with fast swimming. The alpha diversity (Chao1 and Faiths PD) and microbial biomass from the gill of oceanic, neritic fishes were negatively associated with fish mass and a

pelagic lifestyle. We hypothesize that high performance swimming fishes may have adaptations related to keeping gills clear of microbial fouling to maintain respiratory performance. Previous research on Thunnus maccoyii, Southern Bluefin Tuna, has shown that parasitic infection with blood flukes, Cardicola spp., results in pathogenesis of fish from both fluke and egg infections of the gill[52]. Elevated densities of parasite eggs in the gills leads to gill tissue damage, hypoxia and mortality[53]. High acceleration fishes (higher dorsal length to total length ratio), had more sea water associated microbes contributing to the gill, skin, and hindgut as compared to sediment microbes while fishes with a higher microbial biomass on the gills were associated with a higher proportion of sediment sourced microbes. Taken together, we hypothesize that fishes that live closer to shore, such as in the intertidal or subtidal zones as opposed to pelagic fishes, will have lower physiological requirements for swim performance and these fishes will have higher microbial biomass accumulating in the gills as a result of their closer proximity to, or contact with, marine sediments. Gills are the source of both gas exchange and nitrogenous waste excretion in fishes and are composed of a generalized conserved morphology with gill arches, filaments, and lamellae. Understanding how microbes may enhance or disrupt these physiological processes will be important areas of research in the future; especially as it relates to aquaculture production of pelagics[54]. The prevalence of microbiota in the gills across many species of fish is an exciting area to pursue additional research from the perspective of both the host and the microbiota.

Contrary to expectations, we did not observe a direct association between trophic level and alpha diversity. Previous work in mammals and fishes has shown that broad trophic levels are generally associated with hindgut microbiota diversity[15,25,55]. In temperate marine ecosystems, herbivorous marine fishes are rare and therefore it is possible that our limited sampling of the lower trophic extremes could have led to a lack of signal in our study. Our analysis did however show that beta diversity between the midgut and hindgut was generally smaller (more similar) in fishes of lower trophic level with elongated guts. This would suggest that microbial differentiation is greater at the proximal, as opposed to the distal, end of the gut in more carnivorous fishes with shorter intestinal tracts. Although we did not measure stomach content or relative intestinal content, it is possible that the higher trophic fishes have lower feeding frequencies and thus higher rates of fasting in the wild which has been shown to be a strong predictor of gut microbial communities[56]. It is also possible that herbivorous fishes, which feed at a higher frequency rate[57], may have overall more similar microbial communities at the proximal and distal ends of the gut for enhanced nutrient digestion[58]. Our study did attempt to collect mostly adult sized fish but it is possible that age could be a confounding variable as herbivorous fish when juveniles are known to have a higher trophic diet[59]. Future work on associations of alpha diversity and environmental signals such as habitat or fish physiology like swim performance should strive to include a larger replication of species across distant phylogenetic groups to avoid potential biases of phylogenetic pseudo-replication. For instance, although we observe decreasing microbial biomass in the gills of pelagic fishes, it is possible this signal is driven by some evolutionary adaptation which was developed in Scombrids but not in other pelagics.

Phylosymbiosis occurs when the "microbiomes recapitulate the phylogeny of the host" and is primarily studied in guts of invertebrates and mammals[60]. Our study showed that the hindgut, gill, and skin microbiota are more similar in fishes that are more genetically similar. To our knowledge, this is a unique study in vertebrates to evaluate and show phylosymbiosis, in the context of branching length, occurring across multiple body sites. In addition, the discovery of possible phylosymbiosis occurring in the fish gill has not been previously shown. A positive association between the microbiota and host phylogeny is an important finding for guiding future probiotic discovery as most current probiotics used in aquaculture are derived from terrestrial livestock. For vertebrates, phylosymbiosis has primarily been investigated in mammals with a focus on the "internal" gut microbiota[60–62]. Only a few studies have investigated phylosymbiosis on animal surfaces such as mammal skin[63] (38 species, 10 orders)[16], bird feathers (7 species, 1 order)[64], Chondrichthyes vs. Osteichthyes skin (9 species, 9 orders)[65], tropical reef fish skin (44 species, 5 orders)[66]. The latter tropical reef fish skin study used a variety of down sampling methods but the magnitude of association was similar to our study ($p < 0.05$, $r = 0.01$; $p = 0.04$, $r = 0.13$; $p = 0.03$, $r = 0.2$), albeit a different distance measured. These studies sometimes make the error of pseudoreplication by including biological replicates from the same species. It is likely that previous attempts to evaluate phylosymbiosis in fishes have been limited due to limited sampling across evolutionary time scales. In addition, since habitat is an important driver of the fish microbiota, it is important to account for this by having enough samples across diverse habitats as well. For fish, our results suggest that phylosymbiosis is strongest in the hindgut followed by the gill for weighted measures. For gut comparisons, future studies should aim to investigate the importance of reproductive strategies in fishes (viviparity "internal fertilization and live birth" vs. ovoviviparity "internal fertilization" vs. oviparity "external fertilization") to determine if phylosymbiosis and potentially co-phylogeny is stronger in fishes which utilize viviparity. Another important aspect to focus on in future analyses is how microbial diversity corresponds with hosts with high species radiation but shallow overall branching length "low genetic divergence" such as some freshwater cichlids[67]. With a large and recent radiation, it may be possible to further tease apart and test these hypotheses regarding the mechanisms around stochastic microbial exposure leading to potential adaptations in the host leading to co-phylogeny. In light of the observation of possible phylosymbiosis occurring across multiple body sites, we encourage probiotic researchers to consider this when looking for new candidate strains for aquaculture. For example, we show that both Bacillus and Lactobacillus are present across multiple species and body sites, but it is possible that other, unexplored microbial taxa may be effective candidates. Datasets like this can be used to mine and identify where new strains might be residing on the hosts, although future work is necessary to define the functional contributions of fish-associated microbiota using metagenomic and metatranscriptomics methods in addition to cultivation campaigns.

Microbial source tracking analyses showed that sea water contributes more microbes to the fish mucosal environment as compared to sediment. Across body sites, midgut overall had the most microbial sources identified whereas the hindgut and gill had the least (most unknowns). For all body sites however, the majority of microbes were of unknown origin which suggests that further research needs to be conducted to establish a more complete microbial library of the entire marine ecosystem from this California Current Ecosystem "CCE" region where these fishes were obtained. It is also possible that these microbes are simply endemic to the body sites of the fishes from this study, which highlights the importance of integrating concepts of microbial ecology into conservation biology. That is, if a fish goes extinct, so may the microbiome associated with that species of fish. As animal species go extinct, it is possible their microbiota will also follow. Prey items also contribute to the microbial diversity in fishes[68]. A large-scale marine microbial sampling effort should include diverse sediment types, seawater from bays and offshore environments, as well as representatives from the thousands of marine invertebrates, including plankton, and macroalgae species.

Understanding the factors that shape the microbial ecology across vertebrates remains challenging. Our study showed that amongst vertebrates, fishes have the most unique assemblage of total microbial diversity, which we hypothesized corresponds to the deep evolutionary history and habitat types occupied by fishes. An

alternative hypothesis is that more evolutionary ancient lineages of animals would have had more time (generations) to optimize and reduce their microbiome complexity. For gut samples, 92% of ASVs found in fishes were not found in other classes such as mammals, birds, reptiles, and amphibians despite mammals and reptiles having higher gamma diversity at a 50 species cross-section. One caveat of these analyses was that all of our fish samples were wild, whereas many of the mammal, bird, reptile, and amphibian samples were from captive collections[7]. Zoo and wild samples may differ in diversity due to restrictions in diet as well as feeding frequency[14]. In addition, the actual sampling of organisms was not random across the tree but instead was opportunistic based on available samples and data. Future studies should revisit these comparisons when higher species representation is obtained especially from wild samples across a large home range. We demonstrated across one of the largest samplings of species of marine fish to date, that mucosal diversity is greater in body sites outside of the hindgut, particularly the midgut, gill, and skin. This is in contrast to mammals and specifically humans that harbor the highest proportion of microbial diversity concentrated in the hindgut "stool"[69]. Few studies look at the foregut of mammals compared to the hindgut making it difficult to speculate how selection may differ across vertebrate classes. For fish however, because of this drastic difference in midgut to hindgut diversity, it is possible that the majority of midgut microbes are simply from diet and ingested seawater, representing a reservoir of microbes for intestinal colonization. Our results suggested that mammals may have expanded diversity in the hindgut as compared to other vertebrates, whereas fishes may have higher cumulative diversity spread across other body sites. One significant caveat to gamma diversity comparisons in vertebrates is the morphological differences in body sites. Fishes and to some extent amphibians have gills whereas mammals, birds, and reptiles have lungs for breathing. External surfaces such as the "skin" also differ widely. The ideal comparison would be to process and extract the entire animal corpus, however, this is often not feasible due to size limitations. In order to do broad gamma diversity comparisons across vertebrates, one may need to focus on conserved body sites including reproductive organs, oral cavity, and distal gut with the understanding that this would be exclude diversity elsewhere. Higher hindgut diversity in mammals could be explained by the higher occurrence of herbivory in mammals[70]. Our study did not include tropical marine fishes from coral reef ecosystems as we concentrated on the CCE. The CCE does have tropical fish, but they are associated with rocky reef benthic habitat. The addition of gut microbiotas from tropical coral reef fishes, which are primarily localized in tropical environments, may influence this outcome as herbivory is higher in those habitats[71]. In mammals, one of the most important drivers of hindgut microbial diversity is the complexity and physiology of the gut, namely if hindgut fermentation occurs[72]. The adaptation of herbivory may have led to expanded physiological and morphological attributes in the foregut and hindgut leading to a novel ecological niche for microbial colonization and symbiosis of algae-degrading and fermentative taxa[8].

Because few vertebrate microbiota studies are inclusive of multiple body sites, it is difficult to compare cumulative gamma diversity of the fishes here to other vertebrates aside from our observation of higher microbial diversity in external body sites such as the gill and skin in fishes. Our attempts to estimate total microbial diversity across body sites extrapolated to 35,000 fish species demonstrates that despite an incredibly rich dataset with a large range of fish diversity, our investigation of microbial diversity in fishes is only scratching the surface. We suspect the expanded diversity in external sites in fishes may be explained by an evolutionary pressure of a high exposure rate to microbes in the aquatic environment. This may have led to the diversification of the immune system including mucosal site specific lymphoid associated tissue and mucus production[73–75]. Microbiota diversity in the hindgut but not external sites is partially associated with immune system complexity of the hosts[28]. A follow up study would be to compare the immune components (e.g., gene expression, protein, or metabolome profiles) across these different body sites (gill, skin, gut) within an individual fish to determine the extent the host immune system influences (permits or excludes) microbial diversity at a local body site level. In addition, fishes that are naturally exposed to higher microbial diversity in their life history (oligotrophic vs. eutrophic, pelagic vs. benthic) may exhibit differences in the evolution of their immune system.

To fully measure and evaluate microbial diversity in vertebrates, the current research shows it is imperative to design studies that include microbiota samples from a broad phylogenetic sampling of hosts along with multiple body sites. In addition, we demonstrated the importance of including microbial biomass measurements in the context of diversity estimates. Our method to interpolate and approximate microbial biomass of fish mucosal microbiota sites from standards is an important improvement to the field that should be applicable across diverse sample types. By showing phylosymbiosis patterns across multiple body sites, future work could focus on leveraging these findings for probiotic discovery. This study utilized a single fish per species. Based on previous work in Scomber japonicus, Pacific Chub Mackerel, we acknowledge that both age and season will influence the microbial communities of fishes[10]. Thus, we regard this as a foundational study that can serve as a "reference marine fish microbiota". Just as pan-genomes are important to describe genetic variation in Eukaryotes, we can expect microbiota research to follow. Future studies should focus on both obtaining more species representation across the fish tree of life with biological replications within a species, including temporal sampling to constrain seasonality effects, and account for developmental stage. Finally, we make the case for the fish genome community to come together with the fish microbiota community to unify efforts, resources, and data sharing to accelerate discoveries. Testing hypotheses around host-microbiota interactions will only be feasible with more comparative genomics studies and high quality reference genomes of fishes to complement our evolving appreciation of the fish microbiota.

## Methods

### Fish microbiota project (FMP) metadata

**Sampling.** All research complies with relevant ethical regulations (California Department of Fish and Game 2016 Scientific Collecting permit DFW 1379: DocID: D-0018712881-8). A total of 101 fish species were collected from Southern California, primarily in San Diego County, USA ranging from latitude (31.435833 to 33.142589) and longitude (−118.20833 to −117.20879). Fishes were collected using a variety of methods but primarily hook and line, spear, or trawls (for deep-sea fishes). With the exception of the thresher shark and seven gill, all samples (whole fish) were wrapped in aluminum foil, placed in a zip lock bag and then immediately stored on dry ice for ~1–2 h until final storage at −80 °C. For dissections, fishes were allowed to partially thaw and then dissected in batches. The mentioned sharks were dissected immediately and body sites frozen on dry ice followed by storage in −80 °C freezer[76]. For fishes collected by spearfishing, a tandem kayaker was present with a cooler with dry ice. For the deep-sea fishes, once they hit the deck of the ship, there was ~45 min where the fish were being identified until they could be placed on dry ice. Four body sites were processed for microbiota analyses including the gill whole tissue, skin mucus, midgut digesta and hindgut digesta. Each tissue sample was weighed out by weighing the 2 ml extraction tube (DNeasy PowerSoil Pro kit, Cat # 47016) without and then with the sample added (see processing section for more information). For gill samples, in most instances, the whole gill (left 2nd filament) was used. When gills were too large, three slivers of the entire filament were collected from the top, middle and bottom of the gill arch. Skin samples were primarily from scraped mucus. Midgut and hindgut were digesta

material either posterior of the stomach at the beginning of the GI (midgut) or at the anus (hindgut). Details of all fishes used in the study along with its corresponding metadata can be found in (Supplementary Dataset 1 and 2) with the corresponding details of that metadata in the data dictionary (Supplementary Dataset 3). Taxonomy assignments were done using fishtreeoflife.org and fishbase.org, and ultimately NCBI taxonomy[77]. Metadata relating to the various life history features including habitat, biometrics, and diet were summarized from various sources[78,79]. The full data dictionary which describes each metadata column and its subsequent contents is included as a supplement (Supplementary Dataset 3). Geographic information of sampling is presented using ArcGIS free version. Outlines of the fish was obtained using phylopic.org.

**Trophic assignment.** We estimated trophic level by two primary methods. First, we estimated the trophic level by using previously documented diet data derived from the literature. Diet preferences of each species was determined through literature searches including secondary literature databases such as fishbase.org. Where diets differed for juvenile and adult stages, all components were included in the metadata. A coarse estimate of trophic level was determined based on the general diet. If a fish rarely eats fish or if fish are a minor component of the diet, they are still considered in the trophic numbering where 1 = herbivore (consumes primary producers), 2 = primary carnivore (consumes primary consumers such as zooplankton or other invertebrates), 3 = secondary carnivore (consumes small fishes), 4 = tertiary carnivore (consumes large carnivorous fishes), or 5 = quaternary carnivore (consumes high trophic fish or marine mammals). These values can be found in the metadata column (trophic_likely). For the second method, we used the ratio of "relative intestinal length: total body length <RIL:TL>" as an indicator of trophic level[80]. Fishes that are more herbivorous will have a higher RIL greater than 1 and upwards of 5–30 whereas carnivorous fishes will generally be much lower <1[29].

**Reproduction.** Fish are classified in their reproduction method (oviparous = external egg fertilization, ovoviviparous = internal fertilization nourished by egg yolk with live birth, viviparous = internal fertilization nourished by mother gas exchange with live birth) in the metadata column (Reproductive_process).

**Habitat.** Multiple classification methods were used to capture the habitat diversity. The actual minimum and maximum depths are included as (depth_low and depth_high). From here, a broad classification of either shallow, midwater, or deep was used to indicate fishes dwelling between (0–200, 200–1000 m, and <1000 m) (metadata column = habitat_depth_level1). In the next level (metadata column = habitat_depth_level2), we segregate shallow by either intertidal (0–-10 m) or neritic (0–200 m). Fishes dwelling primarily between 200–1000 m are indicated as mesopelagic. Fishes which dwell between 200–1000 m yet are largely demersal are further classified as mesopelagic/benthopelagic. Fishes living between 1000–4000 m are labeled bathypelagic and lastly if dwelling <4000 m are abyssopelagic. Note, the classifications related to habitat depth are based on where the fish primarily reside rather than at what depth they were caught as most have large ranges for vertical migration throughout the day. For instance, most of the deep-sea fishes (classified as bathypelagic) were caught at 500 m. As for salinity tolerance (metadata column = salinity_tolerance) fishes which can live in estuaries are labeled as "brackish" and all others as "marine". Since some fishes undergo greater migrations, we have added a column to indicate (salinity_tolerance_migrations). Fish are either marine (if only in ocean), oceanodromous (migrate long distances in ocean), brackish (spend part of all of life in brackish or estuarine waters),

anadromous, and catadromous. The ocean basin from which fish were caught is indicated by (ocean_basin).

**Swim performance.** Swim acceleration, swim endurance, and the "dorsal length to total length (TL)" ratio are all measures of swim performance. For swim acceleration and endurance, we assigned values to each fish based on the body shape morphometrics previously described for the fish[81]. For acceleration and endurance we assigned either a 1, 2, or 3 with the acceleration value of 3 indicative of high speed (e.g., a barracuda) whereas a high endurance value (3) is indicative of high endurance. In addition, we can describe the capacity for fast swimming based on the placement of the dorsal fin on the fish body. A more forward dorsal fin will be associated with slower swimming fishes whereas fast swimming fishes will generally have a dorsal fin more toward the tail. The dorsal length to TL ratio is a morphometric measure with a higher ratio being indicative of fast swimming or ability to accelerate quickly.

### Microbiota processing
**Fish microbiota project samples.** The fish microbiota project data is held in Qiita study ID 13414[82]. For each extraction set of 96 samples, a set of 8 positive controls were included. These positives were from either the Zymo mock community (Zymo Cat# D6300) or single microbial isolates with known cell concentration based on plate counts. Each isolate or mock was then processed by a serial dilution to extinction to get a range of biomass as input following the KatharoSeq protocol[33,83]. Following the Earth Microbiome Project protocols[6], all DNA extractions were processed using the Qiagen PowerMag kit (Qiagen Mag Attract PowerSoil DNA, Cat # 27100-4-EP) with a modification in that the lysis step was performed in single 2 ml tubes (Qiagen DNeasy PowerSoil Pro kit, Cat # 47016) while the cleanup performed on the KingFisher Flex with Deepwell Head (Thermo Fisher Cat # 5400630) robots using the magnetic bead cleanup. This modification was done to reduce well to well contamination which is common when doing plate–based lysis using vortexing for DNA extractions[84]. In addition, fish species were randomly assigned across the plate. All gDNA was eluted to 60 ul elution buffer included in the kit. For the PCR step, a total of 5 ul of DNA was amplified in a miniaturized 10 ul PCR reaction (Thermo Fisher Platinum Hot Start PCR Mastermix, Cat # 13000014) using the standard EMP protocols for the V4 region of the 16S rRNA gene 515F/806 Rb[85,86]. After PCR, an equal volume of DNA was pooled from all libraries into a single sequencing pool following the KatharoSeq protocol. Equal volume pooling is essential to enable downstream quantification across libraries since sequencing read counts correlate to original DNA input[33,83] to the PCR and subsequently cell biomass to extraction. The final sequencing library was then cleaned up using 1x AMPure XP beads (Beckman Coulter, Cat # A63880) to remove PCR contaminants. Samples were run on three separate sequencing instruments including: MiSeq Nano 2 × 250 bp (artifact ID 102012), NovaSeq 2 × 250 bp (Lane 1 artifact ID 112123, Lane 2 artifact ID 112121), and a MiSeq 2 × 150 bp (artifact ID 113069). All samples were processed according to the protocols outlined in the Earth Microbiome Project (earthmicrobiome.org). Samples were uploaded and processed computationally using Qiita[82], trimmed to 150 bp and ASVs generated using Deblur v1.1.0 and only ASVs passing the positive filter step were used[87]. The analysis artifact ID on Qiita is 46238.

**Microbial biomass estimation.** We developed a methodology to enable the estimation of microbial biomass from a sample. First, we processed the controls through the standard KatharoSeq pipeline to determine the limit of detection[33]. Briefly, this requires one to have standards (microbial isolates or mock communities) with known cell concentrations which can be determined using standard plate counts or other methods. A (8, 10-fold) serial dilution of the standards is

performed, and then extracted alongside actual samples, with a minimum of 32 total positives included on an experiment (4 replicates per dilution). It is critical that all samples and standards are processed identically (same elution volume, same volume into PCR, etc.). It is also critical that the actual biomass of the samples of interest are determined (weighed out). After sequencing, one uses the relationship between read counts and known cell counts of the standards to determine the limit of detection "LoD". We used the setting of 0.9 which indicates the number of reads where 90% of sequences from the controls map to the target bacteria (e.g., Bacillus or Paracoccus in this case). One excludes all samples and controls with reads lower than this cutoff. This is described in great detail in the original KatharoSeq publication[10,33]. To estimate the biomass of samples, one then log 10 transforms the read counts and cell counts of the positive controls which pass the LoD. This assumes a single 16S copy per genome, and thus we are estimating the total 16S copies per gram of tissue. The relationship is modeled using linear regression and the slope and y intercept is then used to estimate the "cell counts" of the actual samples with the log 10 read counts used as input in a similar methodology employed by qPCR. One then must account for the amount of DNA volume used in the library prep along with the elution volume used in DNA extraction. Finally, one must also normalize based on the original biomass used in the extraction to get a final estimate of the microbial cells per gram of "e.g., fish tissue". This method with all normalization steps is now available as a Qiime2 plugin "katharoseq". We have included a detailed Standard Operating Procedure (Supplementary Note 1).

### Microbiota analyses

Alpha, beta, and gamma diversity measures are quantitative measures used in ecology to assess biological diversity as a general function of some discrete area[88]. These values were generally calculated in Qiita using Qiime 2[89]. For alpha diversity, we calculated richness (total unique observed ASVs), Shannon, and Faith's Phylogenetic Diversity. For testing categorical variables against alpha diversity measures, we use the nonparametric Kruskal-Wallis test with a Benjamini Hochberg FDR of 0.05. When comparing continuous metadata factors to alpha diversity, we use Spearman correlation. For beta diversity, we used Unweighted UniFrac which focuses on rare taxa along with Weighted normalized UniFrac which is more heavily weighted or influenced by abundant taxa[90–92]. Any novel sequences to this dataset are integrated and inserted into the phylogenetic tree using SEPP[93]. Statistical testing of metadata categories was performed using PERMANOVA with 999 permutations and a significance of 0.001[94]. Gamma diversity was defined by the total sum of diversity in a given unit which could be a class of vertebrates (e.g., mammals, fish, etc.) or within a given species (gill + skin + gut, etc.).

### Quantitative synthesis of gamma diversity across vertebrate classes.

We first performed a quantitative synthesis comparing hindgut microbiota diversity across numerous vertebrate species. Specifically, we utilized data from previously published or available datasets (Qiita Study ID–Artifact ID–European Nucleotide Archive accession: 11721–111895 [ERP109537] "mackerel 1 year"; 13066–87276 "mackerel year 2–3"; 10353–59141 [ERP106745] "Malawi manure"; 13414–102012, 112121, 112123, 113069 "FMP"; 12227–67067, 67063 [ERP120036] "Australia fish"; 11166 [ERP118494]–56540, 82398, 82409, 82395, 82512, 82400, 82965; 11687–85793, 58423 "SD coastal microbiome"; 12769–81577 "SD map and bioreactor"[7,10,95]. Additional Qiita studies (number is Qiita ID) included are the following: 10353 (Malawi manure), VMP 11166, Salmon/SBT/YTK studies 12227, HMP 1927. Only hindgut samples from fish were used initially and only a single replicate per species was used to eliminate pseudoreplication as a confounding factor. All samples were rarefied to 5000 reads to ensure an even comparison of gamma diversity. A total of 73 fish species (out of

~35,000), 216 bird species (out of ~10,000), 208 mammal species (out of ~5400), 52 reptile species (out of ~10,000), and 20 amphibian species (out of ~6000) were included in the analysis representing a total of ~0.86% of the total vertebrate diversity (569 species out of 66,400 total vertebrate species). (Supplementary Dataset 4). The total list of ASVs were tabulated for each class and then visualized using a Venn diagram (Fig. 8a) using http://bioinformatics.psb.ugent.be/webtools/Venn/. For each vertebrate class (fish, mammal, bird, reptile, and amphibian), the total unique ASVs were calculated by taking the total number of ASVs only found in a given class and dividing by the total number of ASVs found within that class. The total ASVs of a class include ASVs shared amongst other classes (Fig. 8b). An example for fish would be (8810 = total unique ASVs only found in fish/9567 = total ASVs found in fish = 92.1%). Cumulative gamma diversity as a function of sequencing additional species was tabulated for all classes specifically for hindgut microbiota (Fig. 8c). In addition, samples from the fish microbiota project "FMP" which has successful sequencing at each of the four body sites were also included as a comparison (n = 68 species). The total unique microbiota diversity across all four body sites were used as the gamma diversity metric in this case with additive unique ASVs calculated for increasing species sequenced. All additional metrics of gamma diversity for the FMP samples (Fig. 8d–f) were calculated using the 68 fish species which had successful sequencing results for all four body sites. To estimate the total number of unique ASVs (sum of unique ASVs across the four major body sites within a fish) which exist across all fish species, we fit a Michaelis-Menten equation to the results from Fig. 8f using Prism v9. This equation will determine the number of fish species at which approximately half of the microbial diversity will be found (Km value). In addition, the model will produce the maximum y value (Vmax value) which in this example would be the total number or maximum number of unique microbial ASVs. This extrapolation is an estimate which should be updated as more data is added through future experiments or meta-analyses. Cross vertebrate fecal analysis (Lactobacillus discovery) [Qiita analysis 46249].

### Fish microbiota project analysis.

For the FMP samples, samples with less than 1150 reads were excluded leaving a total of 373 successful samples. For the unrarefied raw table, this included 55,069 ASVs which included 1165 chloroplast ASVs which were then removed. For the rarified table (1150 reads), a total of 22,605 ASVs passed filter including 562 chloroplast–associated ASVs which were then removed. Mitochondria reads are removed during the positive filtering step in deblur[87].

### Discovery of probiotic candidates.

The majority of probiotics used in aquaculture are derived from terrestrial sources. Potential bacterial probiotics included any ASV within the following genera: Bacillus, Lactobacillus, Enterococcus, and Carnobacterium (others are Lactococcus, and Weissella) as those have been shown to have a beneficial role as immunostimulants or growth promoters[96]. For our analysis, we focused on Bacillus and Lactobacillus as they are the most commonly used in industrial applications. We identified the prevalence and estimated the relative abundance of these two genera across the FMP dataset.

### Phylogenetic analysis.

Phylosymbiosis can be generally estimated by comparing the phylogenetic distances of the host (in this case fish) to the microbiota similarities of these hosts[60]. To compare the effect of host phylogeny on the microbiota, we generated a phylogenetic tree of all of the fish species used in this dataset using timetree.org[97]. We then used the estimates of evolutionary distance (divergence time) for each pairwise comparison of hosts and tested for associations between host divergence time and gut microbiota divergence using Mantel tests (mantel.rtest function of the R ade4 package). We performed this test for each body site uniquely and determined significance at $p < 0.05$

using a FDR of 0.05. For the microbiota similarity metrics, we included Unweighted UniFrac and Generalized Weighted UniFrac distances.

**Reporting summary**

Further information on research design is available in the Nature Portfolio Reporting Summary linked to this article.

## Data availability

All samples from the Fish Microbiome Project dataset are publicly available since 2022-07-15 at the European Nucleotide Archive under project number PRJEB54736. It is also publicly available on Qiita (Qiita ID 13414). All data used in figures along with code for alpha diversity and phylosymbiosis analyses can be found at: https://github.com/jminich444/Fish_Microbiome_Project. The primary FMP101 (Fish Microbiome Project 101 species) analysis was generated from Qiita analysis ID: 46251 "FMP 2021-07-14_v3 deblur" and is public on Qiita (Supplementary Dataset 6). This includes the following artifacts: 102012, 112121, 112123, and 113069 from Qiita study ID 13414. The read count threshold "limit of detection" and microbial biomass estimate was generated using Qiita analysis ID 49571 and is publically available. The probiotic analysis where *Lactobacillus* and *Bacillus* ASVs were tabulated across vertebrates can be found in Qiita analysis ID 46249 "FMP: broad gamma vertebrate analysis" and is public. Sourectracker2 analysis used BIOM tables generated from Qiita analysis ID: 49005 "FMP101_SDcoastal_[ST2]". This included the following studies: <Qiita study ID 12769; artifact ID 132396> "San Diego Coastal Microbiome Map" and <Qiita study ID 13414; artifact IDs 135520, 135536, 135719> "Fish Microbiome Project". This is included as a Supplementary File (Supplementary Dataset 5). The gamma diversity analysis of comparing the rarified hindgut samples across 569 vertebrates can be found in Qiita analysis ID 46287 "FMP_ultra metaanalysis". This same analysis ID also contains the analysis of the unrarefied fish gamma diversity. This is included as a Supplementary File (Supplementary Dataset 4). The BIOM tables used to generate the microbial biomass estimates for the FMP101 analysis was from Qiita analysis ID 49571. Source data are provided with this paper.

## Code availability

The protocol to perform the Katharoseq method is found in Supplementary Note 1. The code for alpha diversity and phylosymbiosis analyses can be found at: https://github.com/jminich444/Fish_Microbiome_Project. The code for the katharoseq plugin can be found here https://github.com/biocore/q2-katharoseq. The DOI for the code of the katharoseq plugin is 10.5281/zenodo.7217477.

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

## Acknowledgements

We would like to thank the California Department of Fish and Game for permitting this project (2016 Scientific Collecting permit DFW 1379: DocID: D-0018712881-8). We thank Mike Shane and Brice Semmens along with their corresponding organizations for enabling us to collect opportunistic samples which would have otherwise been discarded such as the Hubbs-SeaWorld Research Institute and the California Cooperative Oceanic Fisheries Investigations. We thank the crew of the R/V Robert Gordon Sproul including Phil Zerofski. We especially thank all of the fishers who donated their fish for analysis. We thank Gail Ackermann for help on data submission and metadata. We thank Phil Hastings for insights on fish behavior and ecology. J.J.M. was funded by the NSF Postdoctoral Fellowship in Biology "Rules of Life" Award #2011004. A.H. was supported by funding from the Deutsche Forschungsgemeinschaft (DFG, German Research Foundation)—project number 458274593. This work was supported by grants NSF (OCE-1313747 and EF-2025217) and NIH NIEHS (P01-ES021921) to E.E.A.

## Author contributions

J.J.M. designed the experiment, collected samples, processed samples, curated metadata, analyzed data, and wrote and revised the manuscript. A.H. performed the phylogenetic (phylosymbiosis) analysis and helped write and revised the manuscript. J.V. helped process samples and helped curate metadata, specifically the swim mode data. B.W.F. and Z.R.S. helped in collecting samples and identifying fish species along with reviewing and revising the manuscript. E.K. helped in sample collection and sample processing along with reviewing and revising the manuscript. A.G., D.M., and D.S.P. created the Qiime2 plugin. M.S. helped in sample collection. E.E.A., R.K., and T.P.M. assisted in oversight, mentoring, writing, and revising the manuscript.

## Competing interests

The authors declare no competing interests.
