## [Peer Review File · Nature Communications]

Host biology, ecology and the environment influence microbial biomass and diversity in 101 marine fish speciesReviewer #1 (Remarks to the Author):

The paper by Minich et al. is a major and finely performed contribution in wild/wild caught fish microbial topography. Moreover, the steps they followed in data analysis provides a working framework for any comparative microbiome approach for other animals (maybe plants too!). I have only the following minor comments.

L. 125: a reference is needed here. If you use rrnDB, this database gives 9.7 ± 1.2 16S rRNA gene copy numbers based on 168 genomes

L. 564 Regarding passive collection of samples, details on fish preservation, time after death/capture are needed., e.g. what was the range in time from fish catching to the -80oC

A final comment by the authors is needed on the effect of replicates. Although it is almost impossible to have true replicate samples from the wild, a brief discussion is needed on the validity of the findings in the framework of whether the results can be generalized in the species studied, and the same also applies for the different developmental stages or ages of the fish. Comparing microbiomes of different fish species on individuals who are on different developmental stages and/or age (relative to each of the species life cycle length) might impose some ambiguity in the interpretation of the results.

Reviewer #2 (Remarks to the Author):

The authors characterize the gut microbiota composition of 101 species coming from South California bay and the East coast. The authors did an enormous amount of work collecting all these species, and the study has some novelty in the sense that the authors quantify the microbial biomass, and not only the taxonomic composition of the microbiota in the different body sites, while including a wide collection of species, from different life traits and trophic levels. Moreover, they explain the microbial diversity among vertebrates using the concept of phyllosymbiosis, which has been only described in a few fish studies so far.

Although the study presents interesting results, several points require attention. Starting with the title of the publication, it seems a bit misleading, because first, the authors did not look only at mucosal microbial communities, but also digesta, and especially this makes a lot of difference when looking at the gut. Second, the use of the word microbiome is associated more with the collection of microbes and their genes, while the authors here looked mainly at the compositions of the microbes and how these changes between species. So the more appropriate term would be microbiota and not microbiomes.

In relation to the digesta, the authors, when discussing diversity they should take into account that mucosa can be very different from digesta in terms of microbiota diversity and also how these relate to microbial biomass. So this aspect should be taken into account, especially when claiming that the midgut has the highest diversity compared to the other body sites, but also when similarities are compared between the different gut parts. This may relate to feed microbial-rich components, and not mucosal-associated communities per se.

The authors conclude that body sites were the strongest predictor in affecting the microbial compositions in fish. This is less surprising as this has been reported in several studies before, showing distinct communities between the different body sites (Ruiz-Rodríguez et al., 2020; Lowrey et al., 2015). Another remark that the authors should pay attention to is their discussion concerning microbial diversity. They comment that gill diversity decreases with body size because fish may need to keep the gills 'clean'. However, diversity does not mean that there are no microbes, but it may mean that fish grow and the gill surface area becomes smaller; they may be more dependent on specific microbes that perform nitrogen conversion and gas exchange, which means selected microbes may exist.

Concerning the phylosymbiosis, I am surprised that an r of 0.1 or 0.2 showed a significant correlation, while previous studies have shown that it is required an r of more than 0.2 achieves significant results – see Escalas et al., 2021, Chiarello et al., 2018. Moreover, I think the presentation of the results should be compared to the phylogenetic distance or dissimilarity of the host and not on the divergence in time, as it is very difficult to draw conclusions looking at these figures.

Finally, I am missing more associations and connections between all these species, like the core microbiome, associated with different habitats or body sites and how these various parameters may explain the abundances of specific microbial groups. I can understand that this may be challenging when you have so many species, but this would be interesting to evaluate.

Minor comments:

The authors report that the fish were kept on dry ice or at -80°C until dissection. Especially for the skin and gills, how can the authors make sure that these samples were not affected when placed on dry ice?

Please indicate in the figures what each abbreviation means. It was very difficult to understand what I saw in each figure. The legends should be well explained. Please revise all figures.

Reviewer #3 (Remarks to the Author):

In this manuscript, the authors present a new approach to measure absolute microbial DNA concentration and apply it to a new large DNA amplicon dataset of fish microbiome.

****Short summary****

Data

The new dataset mostly originates from the west coast of the USA and is made of microbial DNA sequences (partial 16S amplicon) from 4 fish body parts across 101 fish species.

Approach

With this dataset and a new DNA quantification method, the authors use a comparative approach to explore the host and environmental factors that are correlated to three aspect of microbial diversity structure within fishes (absolute abundance, richness, composition). The also compare fish microbiome composition to surrounding microbiome composition (SourceTracker analysis) and fish microbiome diversity to other vertebrate classes.

Main findings

- 1/ Microbiome richness and composition but not abundance (as measured by their new proposed technique) differs between host body parts.
- 2/ To a lesser extent, richness, composition and abundance depends on host factors (e.g. phylogeny) and non-host factors (e.g. habitat)
- 3/ Fish microbiome is distinct from other vertebrate classes and other non-host marine environments.

****My main comments****

Form could be improved

I had a hard time not being lost in this manuscript, even if the subject is exciting. I think this because of 1/ the vast amount of results presented: the result section contains about 3500 words and 2/ the lack of clear hypothesis being tested: the introduction is

very short. There are many interesting ideas and data in this paper (e.g. variation of main environmental microbes between species with different ecology, similarity of gut microbiomes in different sites of the intestinal tract across different trophic levels, etc.). I think the MS could be improved by 1/ better framing the study and the hypothesis being tested and 2/ shortening/streamlining some part of the MS (especially the results, see more specific comments/ideas/advices below).

Data is valuable

In itself, the dataset presented is valuable as it allows to explore the microbiome of a vertebrate group that has not been sampled across many host species at the same time before (but note that multiple smaller scale studies exist in fishes). As a consequence, it has the potential to reveal overlooked microbial diversity associated with fishes, a group of vertebrates that has received less attention than mammals, for example.

The new presented approach could be compared with already established approaches
Microbial biomass quantification is a key but overlooked component of the description of microbial community structure, so I think the effort made by the authors to develop a simple technique to measure it is very valuable for the community. The authors demonstrate that their approach works well on extremely simple microbial "community" (2 species from pure cultures) and apply it to very complex communities (with hundreds of species). To be fully convinced by this method, it would be valuable to compare its outcome to more widely used and accepted techniques of absolute DNA quantification in complex communities, such as qPCR.

Results are not surprising but important

The AmpliconSeq results are not surprising, as different body parts represent very different environments and it is known that multiple host and environmental factors modulate microbiome structure. This analysis extends those findings to an understudied group of vertebrates. The absolute abundance results are more interesting and surprising but lack context (e.g. clear hypothesis in the introduction).

Below, I try to provide some more specific/minor advices to improve this manuscript.

****Specific comments****

Abstract.

L 50-51. "Patterns" repeated twice in the same sentence. Maybe change to "identify factors that correlates with" microbial diversity patterns"?

Introduction

L69. There are debates about this. See papers by Louca et al.

L69. Can you precise if you are talking about microbial cells, "species", ASVs or something else?

L 74-76. Not exclusively, there are an increasing number of studies now including wild animals (e.g. Youngblut et al. papers also include wild animals)

L 79 – L80. Phylosymbiosis is not a theory, it is a pattern: please reformulate.

L87. You mention "ecological" and "biological" here, while L72-73 you use "environmental" and "biological". Are there any differences? If yes, it would be useful to explain what those differences are. If no, maybe consider using a consistent terminology.

L 91-92. This is an interesting idea. But can you develop your rationale here: how more "ancient" groups (I am not even sure what this means) are expected to harbor higher diversity of their associated microbiota?

Methods

L 642. "groups of similar phylogeny" -> "closely related groups" ?

L 647. Ref 11 written twice.

L 653. Earth Microbiome *project* ?

L 658. Maybe use the passive form instead of the active form ("you process").

L 658. Confusing repetition of "standards" in this sentence. Please consider reformulate.

L 667. Not being familiar with KatharoSeq, I am not sure what this 0.9 setting represents. Please consider better explaining this threshold for non-specialist audience.

L 668. Add the reference

L 670 "LoD" is not defined before

L 684-685. Usually, when using species data in correlative analysis, one should check for the presence of phylogenetic signal in the residual of the models (to avoid phylogenetic pseudo-replication). This can be for example estimated with Pagel Lambda. Useful references include Felsenstein, J. Inferring Phylogenies. (2004); Uyeda, J. C., Zenil-Ferguson, R. & Pennell, M. W. Rethinking phylogenetic comparative methods. Syst. Biol. 67, 1091-1109 (2018); Pagel, M. Inferring the historical patterns of biological evolution. Nature 401, 877-884 (1999); Revell, L. J. phytools: an R package for phylogenetic comparative biology (and other things). Methods Ecol. Evol. 3, 217-223 (2012).

L 686. How did you construct the microbial phylogeny used for the Unifrac calculation?

L 707-708. The sentence sounds truncated.

L 723-725. I am confused by these dataset references. What are they referring to?

L 729. What about mitochondria? They are also usually removed (host origin).

L 756 - I really appreciate that the authors made available the code and will release the data on EBI. However, I had a hard time navigating the code folder. Would you be able to produce a README file or something similar to explain the content of each folder/scripts deposited on GitHub?

Results

***Biomass estimation. Overall, this section is long, seems repetitive sometimes but is relatively obscure for a non-specialist.**

L 128. What is the "m" stat?

L 120 and L128-129 and L 133-134 seems to be redundant. Consider reformulate or explain why these two statements are not redundant

L 137-157. This could maybe be shortened and included later in the text? E.g. along with the corresponding results. Those are controls to show your results are robust to different technical factors, but we have not read yet the results. Maybe consider reorganizing (this is only my personal opinion).

***Alpha**

L160. Seems like you also include results for microbial biomass estimation. This could be acknowledged in the title of this section, for clarity.

***Beta diversity and phyllosymbiosis**

L 240. Typo: "Body sitel"

L 241-243. This is a very interesting test (Fig. 5b), but it was not really put in context in the introduction.

L 250-L255. You might be able to reduce the length here.

L261: Prediction -> "predictive ability"

L 267. Do you have an empirical reference for co-evolution?

L 268-269. What do you mean by "higher performance" ?

L 267-278. While interesting, this section on probiotics is rather enigmatic to me. It is not really introduced or justified anywhere. I would consider removing it to keep a clearer focus.

***Source pools**

L313-315. Very interesting hypothesis. Can you more explicitly say if you refute it or not?

***Gamma diversity**

L 325-326. Can you explicit a bit more your hypothesis here? Why is it that an "older evolutionary history" implies more gamma diversity?

L 364. I am not sure to understand which model you are referring to here. How this model was constructed? The associated figure shows a Michaelis-Menten model, but I don't find it in the methods. This needs a better explanation.

L 363-364. I don't know what an "old" fish could be, and I don't understand why you ordered the species like that.

L 375-376. Can you explain how microbial biomass estimations feeds into your estimate of fish gamma diversity?

****Discussion**

L 401-402. Can you speculate / give hypothesis why this could be the case?

L407. is microbial fouling a known problem to maintain respiratory performance? Can you provide any reference?

L 414-415. Can you tell us more how this impact the interpretation of your results?

L 433-438. These seem more related to the methods. If you need to save space, this could be shortened.

L 418. Do you mean an evolutionary role of the gill microbiome for its host? Please clarify.

L 471-474. Very interesting idea: can you elaborate a bit more on it?

L 473. By "co evolution", do you mean "co phylogenies"?

L 474-476. Can you explain why this might be important?

L 482-483. Can it be that those microbes are endemic to the fish species you are studying, so there are no "external" sources to look for?

L 491. What do you mean by "diverse evolutionary history"? Could you be more specific?

***Figures**

L1039. Evolutionary divergence is a DISsimilarity (not a similarity). Please clarify.

Fig 6 c and d seems to have the same Y axis label. Please correct

Reviewer #4 (Remarks to the Author):

This study presents an impressively large data set with a diverse composition of marine fish species. However, while the work behind generating this data set and the high number of species included is impressive, I miss a more concrete and underlying hypothesis to be tested. Hence, this manuscript comes across as a very descriptive study

that uses an impressively large data set to draw a range of 'random' correlations based on available data rather than more specific hypothesis driven analyses. This way, the discussion also often seems to just repeat the results section rather than highlighting the broader relevance of truly novel findings.

For example, that species, body site and diet all modify microbiomes is not surprising given the existing literature; so still little is revealed as to what are the functional and evolutionary consequences of these observations among body sites etc.?

Another example is Figure 3 which is kind of underwhelming and not very surprising as some correlations would always be expected with so many samples. That diversity varies among body sites, species and environmental conditions is not very surprising either; contrary what is the relative effect of body site compared to other explanatory variables?

In addition, crucial information relating to the data generation is completely lacking including what microbial DNA markers were amplified and how ASVs were defined (see comments below).

Overall, my feeling is that despite the large data set in terms of number of species, the DNA analyses lack critical information and little new information is revealed in terms of fx the functional and evolutionary role of fish microbiomes. My concern is also illustrated by the descriptive nature of the three research questions put forward in the introduction, which I comment here (author's text listed in quotes):

Lines 86 – forward:

“This study aimed to answer three main questions to understand the ecological and biological drivers of the fish microbiome:

1) what are the primary factors that influence host-associated microbial communities for marine fishes (e.g. body site location, habitat, trophic level, swimming method, phylogeny, etc.)”

Here, you only draw correlations from available data (i.e. not hypothesis driven) while not including e.g. host genetics in more detail than the phyllosymbiosis tests. This way I do not feel that the study is very conclusive as to the bigger picture of this question. What would be more novel is to sequence some host genomes to dig deeper into understanding what host genes are driving the phyllosymbiosis patterns to better understand how the host constrains microbiomes in relation to environmental variance.

“2) where might these microbes originate (sea water, sediment, host, etc)”

This is an interesting question, but the analysis and results addressing this seem inconclusive when they merge so many different species, as it is not very surprising that the surrounding environment always contributes some microbes, but maybe it is not the important ones? Also, it is not clear if some species in the seawater were derived from fishes and not vice versa.

“3) is microbial diversity greater in fishes compared to other vertebrates considering fishes are evolutionarily more ancient.”

The rationale of this question is also not clear – why would more ancient species have higher microbiome diversity? Couldn't it be the opposite that ancient species have had time to evolve their own genome in a way that more effectively selects for only relevant symbiotic species at the expense of random microbes from the environment?

Thus, I think a more specialised journal would be a better fit for this large and useful data set in its current form as it mainly presents value to the fish community. Alternatively, I advice the authors to analyse the data in a more concrete hypothesis driven manner that e.g. addresses more general evolutionary aspects such as the relative roles of environment vs host genetics in shaping mucosal microbiomes.

Other comments:

Throughout: change "isn't" to "is not" and for similar abbreviations

The method to estimate biomass is interesting, but it is not an essential part of the story being told. Further, there is no information given as to if (and how) the method considers copy number variation of 16S genes among bacterial species assuming the method is based on 16S amplicon sequencing (see next comment).

L.643-648: In this part of the methods, you should also specify what DNA has been amplified in the PCR? I assume it is some type of 16S based primers, but there is no information given regarding what is amplified, and this is one of the most important details for the entire protocol.

L.679: There is also no description on how ASVs were defined, and which similarity thresholds were used?

L.733: You may want to temper this title to discovery of probiotic candidates. Also, the first argument in this paragraph is that probiotics of terrestrial origin are not good representatives for fishes. Then you still decide to only look for groups that are well known probiotics in terrestrial animals; I would question that this is the best way to discover potentially new useful probiotics in fishes?

Our responses to the reviewer comments are detailed below. The line numbers refer to the clean version.

REVIEWER COMMENTS

Reviewer #1 (Remarks to the Author):

The paper by Minich et al. is a major and finely performed contribution in wild/wild caught fish microbial topography. Moreover, the steps they followed in data analysis provides a working framework for any comparative microbiome approach for other animals (maybe plants too!). I have only the following minor comments.

We thank this reviewer for their input and agree 100% this can be applied to plants or any other host-associated or free-living system. In regards to the method of microbial biomass estimation, we have added a paragraph in the discussion section (Line 456-481): “One of the biggest challenges in microbial ecology...” to drive home this point. Also we list several sample types this method can be applied to (Line 479) “We leverage this observation and provide a framework for consistent sample processing to enable insights into microbial biomass which should be applicable across all ecosystems including animals, plants, soil, water, and the air.”

L. 125: a reference is needed here. If you use rrnDB, this database gives 9.7 ± 1.2 16S rRNA gene copy numbers based on 168 genomes

Thank you for this comment. We understand this challenge of varying 16S copy numbers across microbial species. We also understand this varies wildly across environments and that it will vary depending on the sample type being processed (free living vs host-associated, etc). Because of this variability, we just stuck with the general metric of estimating 16S copies per gram of tissue. We base this decision on a recent benchmarking paper showing that its essentially really hard to do this accurately and any method which attempts to normalize per copy number may be introducing more error [PMID: 29482646]. If a researcher knows their system has on average 4 16S copies per genome, they could make that normalization. We did not want to force this normalization onto people because we recognize that different environments will have

different microbes. Also, many microbes simply don't have full genomes sequenced and thus we don't know copy numbers.

We have added a line in the methods to specify this (Line 831)

"This assumes a single 16S copy per genome, and thus we are estimating the total 16S copies per gram of tissue."

We state this in the discussion (Line 473-477)

"One caveat is that many microbes have multiple 16S copies per genome, thus one may want to perform this normalization on their sample set in the future if rrn copy number is known across the resident microbial taxa. Attempting to normalize per 16S copy number per genome however is challenging and most studies actually fail to do this, thus we have avoided this additional normalization attempt in our workflow⁴⁰"

L. 564 Regarding passive collection of samples, details on fish preservation, time after death/capture are needed., e.g. what was the range in time from fish catching to the -80°C

→ This is a good point. We have clarified our sentences in the methods section.

(Line 711-719)

"With the exception of the thresher shark and seven gill, all samples (whole fish) were wrapped in aluminum foil, placed in a zip lock bag and then immediately stored on dry ice for approximately 1-2 hours until final storage at -80 °C. For dissections, fishes were allowed to partially thaw and then dissected in batches. The mentioned sharks were dissected immediately and body sites frozen on dry ice followed by storage in -80 °C freezer⁷⁷. For fishes collected by spearfishing, a tandem kayaker was present with a cooler with dry ice. For the deep-sea fishes, once they hit the deck of the ship, there was approximately 45 minutes where the fish were being identified until they could be placed on dry ice."

A final comment by the authors is needed on the effect of replicates. Although it is almost impossible to have true replicate samples from the wild, a brief discussion is needed on the validity of the findings in the framework of whether the results can be generalized in the species studied, and the same also applies for the different developmental stages or ages of the fish. Comparing microbiomes of different fish species on individuals who are on different developmental stages and/or age (relative to each of the species life cycle length) might impose some ambiguity in the interpretation of the results.

→ We agree and we've added these caveats to the conclusion.

(Line 688-697)

"This study utilized a single fish per species. Based on previous work in *Scomber japonicus*, Pacific Chub Mackerel, we acknowledge that both age and season will influence the microbial communities of fishes¹⁰. Thus, we regard this as a foundational study that can serve as a 'reference marine fish microbiota'. Just as pan-genomes are important to describe genetic variation in Eukaryotes, we can expect microbiota research to follow. Future studies should focus on both obtaining more species representation across the fish tree of life with biological replications within a species, including temporal sampling to constrain seasonality effects, and account for developmental stage. Finally, we make the case for the fish genome community to come together with the fish microbiota community to unify efforts, resources, and data sharing to accelerate discoveries"

Reviewer #2 (Remarks to the Author):

The authors characterize the gut microbiota composition of 101 species coming from South California bay and the East coast. The authors did an enormous amount of work collecting all these species, and the study has some novelty in the sense that the authors quantify the microbial biomass, and not only the taxonomic composition of the microbiota in the different body sites, while including a wide collection of species, from different life traits and trophic levels. Moreover, they explain the microbial diversity among vertebrates using the concept of phyllosymbiosis, which has been only described in a few fish studies so far.

Although the study presents interesting results, several points require attention.

Starting with the title of the publication, it seems a bit misleading, because first, the authors did not look only at mucosal microbial communities, but also digesta, and especially this makes a lot of difference when looking at the gut.

→ We have removed the word mucosal from the title.

Second, the use of the word microbiome is associated more with the collection of microbes and their genes, while the authors here looked mainly at the compositions of the microbes and how these changes between species. So the more appropriate term would be microbiota and not microbiomes.

→Changed throughout (from microbiome to microbiota)

In relation to the digesta, the authors, when discussing diversity they should take into account that mucosa can be very different from digesta in terms of microbiota diversity and also how these relate to microbial biomass. So this aspect should be taken into account, especially when claiming that the midgut has the highest diversity compared to the other body sites, but also when similarities are compared between the different gut parts. This may relate to feed microbial-rich components, and not mucosal-associated communities per se.

→As recommended previously, we've changed the title of the manuscript, namely removing 'mucosal'. We've updated the description and emphasized 'digesta' in the methods.

(Line 719-725)

“Four body sites were processed for microbiota analyses including the gill whole tissue, skin mucus, midgut digesta and hindgut digesta. Each tissue sample was weighed out by weighing the 2 ml extraction tube without and then with the sample added. For gill samples, in most instances, the whole gill (left 2nd filament) was used. When gills were too large, three slivers of the entire filament were collected from the top, middle and bottom of the gill arch. Skin samples were primarily from scraped mucus. Midgut and hindgut were digesta material either posterior of the stomach at the beginning of the GI (midgut) or at the anus (hindgut). “

The authors conclude that body sites were the strongest predictor in affecting the microbial compositions in fish. This is less surprising as this has been reported in several studies before, showing distinct communities between the different body sites (Ruiz-Rodríguez et al., 2020; Lowrey et al., 2015).

→ We thank the reviewer for this comment. We've added both citations to the discussion. We explain some of the differences though below.

(Line 99-104)

“Body site is frequently one of the strongest predictors when comparing single fish species including Rainbow Trout ²⁰, Atlantic Salmon ²¹, Pacific Chub Mackerel ^{10,21}, Yellowtail Kingfish ²², and Southern Bluefin Tuna ²³. A study which analyzed microbiota from 13 species of fish from 5 families all caught in a bay, also found the body site was a major driver of microbial composition ²⁴. This study however combined the entire gut microbiota from midgut and hindgut.”

Our study differs from the first study (Ruiz-Rodríguez) in that it includes the midgut digesta. They combined the midgut and hindgut (entire GI tract) into a single sample

and thus likely lost a lot of resolution in the spatial gut microbiome. Their study also only had 13 species of fish (compared to 101 species from West Coast and 15 extra species from East Coast). We have almost an order of magnitude more replication. Furthermore, their 13 species are only across 5 families whereas we have representatives from 55 families. One thing to note is that although body site is significant across the large species dataset, this value is weaker when compared to a single species of fish where the prediction is much higher. We have added 2 sentences about the Ruiz-Rodriguez paper into the discussion.

Lowrey 2015 is a great foundational paper to this, but they only are looking at 1 species of fish (rainbow trout). We cite several of the papers where we compared the body site across a single species of fish.

Another remark that the authors should pay attention to is their discussion concerning microbial diversity. They comment that gill diversity decreases with body size because fish may need to keep the gills 'clean'. However, diversity does not mean that there are no microbes, but it may mean that fish grow and the gill surface area becomes smaller; they may be more dependent on specific microbes that perform nitrogen conversion and gas exchange, which means selected microbes may exist.

→ We thank the reviewer for the comment. We believe they were commenting on the discussion section. We realize that the wording was slightly ambiguous and we have clarified (text below). It's also possible there was some confusion so we explain further. There are 2 primary comparisons on this. First, we compare the alpha diversity metrics. In this study we focused on the most commonly used ones in the field (Chao1, Faiths PD, and Shannon). In the second comparison, we compare microbial biomass. Part of the main aspect of this manuscript is developing and using a new method to generate data surrounding. We agree that these are indeed two completely different measurements. Since they are somewhat similar in that they are a single data point for each sample, we decided to aggregate the results together into a single visualization (Figure 3). Figure 4 is a follow up to figure 3 but it is solely looking at a subset of samples (neritic fish collected from the ocean 'bay/estuary' fishes excluded). Figure 4 also focuses on the metrics associated with microbial biomass in the gill.

(Line 525-531)

“We observed decreasing microbial diversity (Chao1 and Faiths PD) in the gills of larger fishes and fishes that were morphologically associated with fast swimming. The alpha diversity (Chao1 and Faiths PD) and microbial biomass from the gill of oceanic, neritic fishes were negatively associated with fish mass and a pelagic lifestyle. We hypothesize that high performance swimming fishes may have adaptations related to keeping gills clear of microbial fouling to maintain respiratory performance.”

Concerning the phylosymbiosis, I am surprised that an r of 0.1 or 0.2 showed a significant correlation, while previous studies have shown that it is required an r of more than 0.2 achieves significant results – see Escalas et al., 2021, Chiarello et al., 2018. Moreover, I think the presentation of the results should be compared to the phylogenetic distance or dissimilarity of the host and not on the divergence in time, as it is very difficult to draw conclusions looking at these figures.

→ Thanks for the comment. For Mantel correlation tests, the P value is what you would look at to determine significance followed by the r value -1 to 1 to establish the magnitude and direction. Moreover, many studies looking at phylosymbiosis utilize divergence time estimates as they do indeed correlated with branch length. Based on the existing literature, divergence time is also more interpretable to a broader audience as compared to other phylogenetic distance measures. The first paper Escalas et al. 2021 only had 12 species and they did not find any significant signal with the Sparidae. They examined the hindgut digesta of these fishes. One of the potential problems with the study design was that fish were caught by gill net and then stored on ice for an extended period of time (3hrs) which could have been enough time for the communities composition to change. Aside from that sampling detail, Sparidae is relatively recent with a common ancestor of around ~60 MYA. This study relied on previous work that used 5 loci across 91 species to get a time-calibrated phylogeny of the species. The second study by Chiarello et al. 2018, looked at skin microbiomes from 44 species of coral reef fish. They only looked at weighted unifrac and did some sort of down-sampling which wasn't exactly clear. Either way they found significant values for phylosymbiosis for weighted unifrac for lower r values ($P < 0.05$, $r = 0.01$; $P = 0.04$, $r = 0.13$; $P = 0.03$, $r = 0.2$). Also, this study used divergence time on the X-axis as well.

We absolutely agree that using multiple loci and even whole genome comparisons of the 116 species in this study would have been ideal, but it would be a significant undertaking and likely a separate manuscript in itself. An alternative to generating massive sequencing data for all of the species of fish is to use timetree which essentially is a large database updated every few years to pull in studies like the Sparidae one and implement divergence times across the Eukaryotic tree of life. Timetree is a program first published in 2006 with the most recent version update in

2017 (1490 citations). It is also commonly used in many published phyllosymbiosis papers (at least 17 from our search). This method yielded an appropriate phylogenetic tree of all the fish used in this study which consisted of some very distant members >400 MYA divergence times.

That all being said, we did attempt to generate tree using existing sequence data. For our 116 species of fish, we did find the CO1 marker gene for all and attempted to create phylogenetic trees and thereby use various phylogenetic distance measures rather than divergence times. We explain our methods below: DNA metabarcoding analyses with the COI marker gene has been used to assess the phylogenetic structures of marine fishes [PMID: 29856794]. Following this approach, we downloaded CO1 sequences ~550-650 bp from all 116 species of fish using the BOLD database. [https://v4.boldsystems.org/index.php/Public_SearchTerms] [PMID: 18784790]. Using the MEGA11 software [PMID: 33892491], first CO1 sequences were aligned using Muscle with default settings [PMID: 15034147]. Pairwise genetic distances between all 116 species were calculated using the Kimura two parameter 'K2P' model [PMID: 7463489] and a phylogenetic tree was created using Neighbor-joining (bootstrapping n=1000). The K2P distance matrix (genetic distance) was then compared against our timetree distance matrix (divergence time). We also attempted creating a tree using another common tool RAxML [PMID: 24451623]. All of the trees estimated tenuous to outlandish relationships with low support, which made them not usable for any comparisons. The trees work great for resolving species within a family but generally fail when trying to implement very distant clades because we are relying on a single, fairly rapidly evolving locus. We emphasize this problem in the discussion and call on collaborations between fish taxonomists, evolutionary biologists, and microbiome researchers to work together.

Finally, I am missing more associations and connections between all these species, like the core microbiome, associated with different habitats or body sites and how these various parameters may explain the abundances of specific microbial groups. I can understand that this may be challenging when you have so many species, but this would be interesting to evaluate.

→ Thank you for the comment. There are many different directions one could take this dataset. As a foundational paper to the field, we hope others will use this dataset in meta-analyses to their datasets to explore some of these questions. With regards to the core microbiome, there are a lot of different ideas about how to do this properly with

very little consensus as to what's 'ideal'. There is a nice perspective on this published last year PMID: 34862327. Because of this, we opted to steer clear of a focus on the core microbiome.

Minor comments:

The authors report that the fish were kept on dry ice or at -80oC until dissection. Especially for the skin and gills, how can the authors make sure that these samples were not affected when placed on dry ice?

→ Thank you for pointing this out. Because of space restrictions, we didn't go into details in the original manuscript but this is a really important point. We've now updated the manuscript methods to clarify how samples were stored. Specifically, we wrapped all fish in foil followed by placement in a zip lock bag. This minimizes/prevents cross contamination.

Please indicate in the figures what each abbreviation means. It was very difficult to understand what I saw in each figure. The legends should be well explained. Please revise all figures.

→ All abbreviations added to the bottom of each figure legend

Figure 1: Added this line at the bottom

"MG=midgut, HG=hindgut, GI=gastrointestinal tract, m=meters"

Figure 2: "Abbreviations: QC = quality control, g__ = refers to a genus of bacteria, HM mock=homemade mock or human made mixture of bacteria to use as a control whereas zymo mock = mock microbial community created by a company 'Zymo'"

Figure 3: We've updated this figure to specify the test statistic for Kruskal-Wallis 'H' to make it more clear. We've also added those details in the abbreviations in the legend. "Abbreviations: KW or KW stat='H' test statistic from Kruskal-Wallis test, MG=midgut, HG=hindgut, Faith PD=Faith's Phylogenetic Diversity metric, GI:TL=gastrointestinal length to fish total length 'ratio', TL=total length of fish"

Figure 4

We've edited the y axis labels for panels "E, F, G, H" for Chao1 and Faiths PD to reflect what was presented on Figure 3 (be consistent) and avoid confusion.

“Abbreviations: habitat_act_collection=refers to the metadata column name from where this habitat classification can be found... SZ=surf zone, RIT=rocky intertidal, RST=rocky subtidal, IS=inner shelf, KBRF = kelp bed rocky reef, MDRF=mid depth rocky reef, CP=coastal pelagic, P=pelagic. Mass_g_log = log 10 (mass of the fish in grams), distance_from_shore_m_log=log 10 (distance from nearest point on shore in meters from where the fish was caught).”

Figure 5

→In legend added

“ b) Impact of trophic level on similarity between midgut and hindgut (within a species) (linear model: P= P value, m=slope, dotted lines are 95% Confidence Interval).”

“Abbreviations: F-Stat = test statistic used in PERMANOVA analysis, all row names in 5a are metadata column names used in the analysis, MG=midgut, HG=hindgut, Gen. Weighted UniFrac = Generalized Weighted UniFrac”

Figure 6

“Abbreviations: * P<0.05, ** P<0.01, *** P<0.001, **** P<0.0001, ASV=amplified sequence variant ~unique sub-Operational Taxonomic Unit, SD=standard deviation, MG=midgut, HG=hindgut, KW=Kruskal-Wallis test statistic '*H*', IQR=inter quartile range, SW=sea water”

Figure 7

“Abbreviations: ASV=amplified sequence variant, 5k=5000”

Reviewer #3 (Remarks to the Author):

In this manuscript, the authors present a new approach to measure absolute microbial DNA concentration and apply it to a new large DNA amplicon dataset of fish microbiome.

****Short summary****

Data

The new dataset mostly originates from the west coast of the USA and is made of microbial DNA sequences (partial 16S amplicon) from 4 fish body parts across 101 fish species.

Approach

With this dataset and a new DNA quantification method, the authors use a comparative approach to explore the host and environmental factors that are correlated to three aspect of microbial diversity structure within fishes (absolute abundance, richness, composition). The also compare fish microbiome composition to surrounding microbiome composition (SourceTracker analysis) and fish microbiome diversity to other vertebrate classes.

Main findings

- 1/ Microbiome richness and composition but not abundance (as measured by their new proposed technique) differs between host body parts.
- 2/ To a lesser extent, richness, composition and abundance depends on host factors (e.g. phylogeny) and non-host factors (e.g. habitat)
- 3/ Fish microbiome is distinct from other vertebrate classes and other non-host marine environments.

****My main comments****

Form could be improved

I had a hard time not being lost in this manuscript, even if the subject is exciting. I think this because of 1/ the vast amount of results presented: the result section contains about 3500 words and 2/ the lack of clear hypothesis being tested: the introduction is very short. There are many interesting ideas and data in this paper (e.g. variation of main environmental microbes between species with different ecology, similarity of gut microbiomes in different sites of the intestinal tract across different trophic levels, etc.). I think the MS could be improved by 1/better framing the study and the hypothesis being

tested and 2/ shortening/streamlining some part of the MS (especially the results, see more specific comments/ideas/advices below).

→ We thank the reviewer for this criticism. We have restructured the manuscript to make it more cohesive including expanding the introduction to better describe our hypotheses and experiment. We have also changed the title to better reflect the manuscript. We have reduced the results where feasible.

Data is valuable

In itself, the dataset presented is valuable as it allows to explore the microbiome of a vertebrate group that has not been sampled across many host species at the same time before (but note that multiple smaller scale studies exist in fishes). As a consequence, it has the potential to reveal overlooked microbial diversity associated with fishes, a group of vertebrates that has received less attention than mammals, for example.

→ Thank you for recognizing the efforts taken for this project and the intrinsic value of the dataset

The new presented approach could be compared with already established approaches

Microbial biomass quantification is a key but overlooked component of the description of microbial community structure, so I think the effort made by the authors to develop a simple technique to measure it is very valuable for the community. The authors demonstrate that their approach works well on extremely simple microbial “community” (2 species from pure cultures) and apply it to very complex communities (with hundreds of species). To be fully convinced by this method, it would be valuable to compare its outcome to more widely used and accepted techniques of absolute DNA quantification in complex communities, such as qPCR.

→ Thank you for this comment. We have added an entire paragraph to the discussion to address this point and our findings in context to other strategies. At the start of this manuscript, there was not a commercially available and highly quantitative mock community (100s of species of bacteria). Of course, one could make their own, but it then becomes very challenging for others to reproduce the results. We certainly believe and hope that companies will start producing more complex mock communities. That being

said, we have published the Kathoroseq protocol (LoD) across several studies. In many of those studies, we use the zymo mock community. We did not use the zymo mock for quantitative purposes here because zymo does not give an accurate concentration of their cells in the mock which is problematic when using a standard curve. We go into this in the manuscript, but qPCR is often incorrectly used in microbial quantitation. If you are using a 16S type gene target, most researchers don't realize that you will amplify mtDNA from the host. A qPCR method works great in a free-living community but this is pretty rare when it comes to actual research projects. You can use a single copy gene as well for qPCR, but then you must start changing the annealing temperature etc and it's really hard to dial in on a repeatable measurement which works well across different microbes. We also discuss the challenges with flow cytometry.

(Line 456-481)

“One of the biggest challenges in microbial ecology is estimating microbial biomass also known as total microbial load or absolute abundance. This is especially challenging in host-associated environments as the existing methods to do this including microscopy, flow cytometry, and qPCR³⁴ are optimized to work best on free-living systems and may have lower throughput or require immediate processing. Flow cytometry methods are challenging due to excessive clumping of cells, lower throughput, and require immediate processing of samples³⁴ while qPCR assays (e.g. 16S) can have many false positives due to contamination and thus co-amplification of either chloroplasts or mitochondria³⁵. Another approach is to use a spike-in of known cell counts of control bacteria³⁶ or synthetic oligos³⁷ into your samples which can enable biomass estimations post sequencing. The downside to this approach is you lose sequencing depth across samples and if using cells, it is necessary to ensure that those microbes are not already found in your samples. Many studies ignore estimating biomass and instead are forced to assume equal biomass across samples. This strategy causes problems when trying to compare relative abundances of microbes across samples which has led to many computational tools being developed around compositionality^{38,39}. Our method is extremely useful and novel in that it enables one to estimate the microbial biomass (namely number of 16S copies per gram of tissue) using standard 16S high throughput sequencing and without contaminating samples with a spike-in. One caveat is that many microbes have multiple 16S copies per genome, thus one may want to perform this normalization on their sample set in the future if *rrn* copy number is known across the resident microbial taxa. Attempting to normalize per 16S copy number per genome however is challenging and most studies actually fail to do this, thus we have avoided this additional normalization attempt in our workflow⁴⁰. Our results directly refute the hypothesis that sequencing depth is random and instead show that it is highly correlated to the microbial

biomass. We leverage this observation and provide a framework for consistent sample processing to enable insights into microbial biomass which should be applicable across all ecosystems including animals, plants, soil, water, and the air.”

Results are not surprising but important

The AmpliconSeq results are not surprising, as different body parts represent very different environments and it is known that multiple host and environmental factors modulate microbiome structure. This analysis extent those finding to an understudied group of vertebrates. The absolute abundance results are more interesting and surprising but lack context (e.g. clear hypothesis in the introduction).

Below, I try to provide some more specific/minor advices to improve this manuscript.

→ We are so thankful for reviewer 3's comments. We have made critical changes to improve the flow and clarify of the manuscript.

****Specific comments****

Abstract.

L 50-51. "Patterns" repeated twice in the same sentence. Maybe change to "identify *factors that correlates with* microbial diversity patterns"?

→ changed (Line 48-51)

"The microbiota from 101 species of Southern California marine fishes, spanning 22 orders, 55 families, and 83 genera representing ~25% of local marine fish diversity, was analyzed to identify factors that explain microbial diversity patterns in a geographical subset of marine fish biodiversity."

Introduction

L69. There are debates about this. See papers by Louca et al.

L69. Can you precise if you are talking about microbial cells, "species", ASVs or something else?

→ We've clarified our statements and have added a line to emphasize that although there might be a huge phylogenetic diversity, many of these microbes share functional/metabolic redundancy.

(Line 71-73)

“The Earth may contain around 10^{30} microbial cells ¹ distributed across 10^{12} species of microbes (Bacteria and Archaea) ², albeit many of these species likely share functional metabolic redundancy ^{3,4}.

L 74-76. Not exclusively, there are an increasing number of studies now including wild animals (e.g. Youngblut et al. papers also include wild animals)

→ We've modified the sentence to clarify that yes some studies are now getting wild samples which is fantastic. We actually cite this paper (Youngblut) in the next sentence.

That being said, the Youngblut study was indeed all stool and they only analyzed 9 species of fish (out of ~36000). Of the 9 fishes, 6 were from the same family (carps). Also they were all from temperate/cold freshwater environments. They did a good job getting wild samples, but this is unacceptable to claim 9 fish samples represent that class.

(Line 78-81)

“Of the large meta-analyses that have sought to evaluate vertebrate host microbial diversity, most focus exclusively on hindgut or stool from terrestrial animals from captive (zoo) environments whereas studies from wild animals may yield different findings ^{7,12,14}.”

L 79 – L80. Phyllosymbiosis is not a theory, it is a pattern: please reformulate.

→ changed

(Line 81-86)

“Fishes, despite being the most phylogenetically diverse vertebrates, are severely underrepresented in these studies ^{7,12}. This underrepresentation is a critical concern because many hypotheses and patterns have arisen from these studies, including phyllosymbiosis and contributions of diet driving community assemblies, yet body sites outside the gut are largely ignored and aquatic animals are insufficiently sampled to establish a generality of conclusions.”

L87. You mention “ecological” and “biological” here, while L72-73 you use “environmental” and “biological”. Are there any differences? If yes, it would be useful to explain what those differences are. If no, maybe consider using a consistent terminology.

→ Good catch. We've updated this. In general environmental refers to environmental impacts/associations with a single species (which often times microbiome studies are on

single host species) whereas ecological would be referencing environmental effects across multiple host species. We've emphasized ecological for this study as we look at multiple species.

(Line 75-78)

"The host-associated gut microbiota in vertebrates is shaped by a variety of biological factors including host phylogeny, diet, and age, along with environmental or ecological factors such as geography, habitat, and climate, whereas less is known about other body sites ^{7,8,10-13}."

(Line 120-121)

"This study aimed to answer three main questions to understand the ecological and biological drivers of the fish microbiota of fishes"

L 91-92. This is an interesting idea. But can you develop your rationale here: how more "ancient" groups (I am not even sure what this means) are expected to harbor higher diversity of their associated microbiota?

→ We've updated this statement/question to be more specific. (Line 126-128)

"is microbial diversity greater in fishes compared to other vertebrates considering fishes are more genetically diverse"

Methods

L 642. "groups of similar phylogeny" -> "closely related groups" ?

→ We just deleted that second part of the sentence because it adds unnecessary information. The fish were randomly placed throughout the plate is good enough here.

(Line 799-800)

"In addition, fish species were randomly assigned across the plate."

L 647. Ref 11 written twice.

→ deleted

L 653. Earth Microbiome *project* ?

→ Edited the sentence to specify

(Line 809-810)

“All samples were processed according to the protocols outlined in the Earth Microbiome Project (earthmicrobiome.org)”

L 658. Maybe use the passive form instead of the active form (“you process”).

→ thanks for the catch, changed to past tense (Line 817)

“First, we processed the controls”

L 658. Confusing repetition of “standards” in this sentence. Please consider reformulate.

→ Thanks, changed first standards to ‘control’s (Line 817-818)

“First, we processed the controls through the standard KatharoSeq pipeline to determine the limit of detection ³³.”

L 667. Not being familiar with KatharoSeq, I am not sure what this 0.9 setting represents. Please consider better explaining this threshold for non-specialist audience.

→ Good point. We generally wanted to avoid excessive words in the first pass but will spend some more time explaining. (Line 826-828)

“We used the setting of 0.9 which indicates the number of reads where 90% of sequences from the controls map to the target bacteria (e.g. *Bacillus* or *Paracoccus* in this case)”

L 668. Add the reference

→added

L 670 “LoD” is not defined before

→ Added this abbreviation above (Line 824-826)

“After sequencing, one uses the relationship between read counts and known cell counts of the standards to determine the limit of detection ‘LoD’.”

L 684-685. Usually, when using species data in correlative analysis, one should check for the presence of phylogenetic signal in the residual of the models (to avoid phylogenetic pseudo-replication). This can be for example estimated with Pagel Lambda. Useful references include Felsenstein, J. *Inferring Phylogenies*. (2004); Uyeda, J. C., Zenil-Ferguson, R. & Pennell, M. W. Rethinking phylogenetic comparative methods. *Syst. Biol.* 67, 1091–1109 (2018); Pagel, M. Inferring the historical patterns of biological evolution. *Nature* 401, 877–884 (1999); Revell, L. J. *phytools*: an R package for phylogenetic comparative biology (and other things). *Methods Ecol. Evol.* 3, 217–223 (2012).

→ Faith's Phylogenetic diversity is an alpha diversity metric which is calculated by summing the branch lengths across all features within a sample. This method is very commonly used in microbial ecology to assess and tabulate the microbial diversity of a sample, reducing it to a single number. This methodology is very useful to the field and in some instances is agnostic to the organisms within the sample. For instance, you could have 2 samples with the same Faith's PD value but with a completely different set of microbes within them. In any rate, because it's a single number, you can apply standard statistical analyses. For instances of determining if microbial diversity differs amongst categorical variables (e.g. fish habitat names: deep sea vs shallow), we use the Kruskal-Wallis test. For instances where we have numeric values as the variables (e.g. length of fish, mass of fish, GI length : Total fish length ratios, etc), we apply Spearman correlation.

Faith DP (1992) Conservation evaluation and phylogenetic diversity. *Biol Conserv* 61:1–10. [https://doi.org/10.1016/0006-3207\(92\)91201-3](https://doi.org/10.1016/0006-3207(92)91201-3)

L 686. How did you construct the microbial phylogeny used for the Unifrac calculation?

→ Here we use UniFrac as implemented within Qiita and Qiime2. The 'SEPP' program is used to integrate/insert any new sequences into the tree PMID: 29719869. **We have updated our methods to reflect this.**

(Line 848-851)

“For beta diversity, we used Unweighted UniFrac which focuses on rare taxa along with Weighted normalized UniFrac which is more heavily weighted or influenced by abundant taxa⁹⁰⁻⁹². Any novel sequences to this dataset are integrated and inserted into the phylogenetic tree using SEPP⁹³.”

UniFrac is a very commonly used tool in microbial ecology to generate B diversity distance matrices. It differentiates itself from other B diversity measures in that it utilizes a phylogenetic tree of the features within that sample. The most common application in microbiome datasets (default) is to use the branch length of the 16S small subunit rRNA gene. The first paper has 6700 citations. Within Qiita, we used

Lozupone, Catherine, and Rob Knight. "UniFrac: a new phylogenetic method for

comparing microbial communities." *Applied and environmental microbiology* 71.12 (2005): 8228-8235.

L 707-708. The sentence sounds truncated.

→ We've deleted this sentence and just reference the supplemental file after that sentence.

L 723-725. I am confused by these dataset references. What are they referring to?

→ Placement here is out of order. We have moved it up. These are all studies used in the large meta-analysis across the vertebrate samples. The Qiita study ID listed refers to the publicly available dataset within Qiita.

Line 864-865

"Additional qiita studies (number is Qiita ID) included are the following: 10353 (malawi manure), VMP 11166, Salmon/SBT/YTK studies 12227, HMP 1927)."

L 729. What about mitochondria? They are also usually removed (host origin).

→ Indeed, thanks for bringing this up - we have updated our methods to clarify that mitochondria are removed during the positive filtering step performed by deblur. PMID: [28289731](https://pubmed.ncbi.nlm.nih.gov/28289731/) (Line 902)

"Mitochondria reads are removed during the positive filtering step in deblur⁸⁷."

L 756 – I really appreciate that the authors made available the code and will release the data on EBI. However, I had a hard time navigating the code folder. Would you be able to produce a README file or something similar to explain the content of each folder/scripts deposited on GitHub?

→ Thanks for this comment. We have created a README for the GitHub along with creating an end-end SOP (standard operating procedure) to enable others to use this method in their own dataset. We have deposited this SOP as a supplement along with on GitHub.

Results

*Biomass estimation. Overall, this section is long, seems repetitive sometimes but is relatively obscure for a non-specialist.

L 128. What is the “m” stat?

→ replaced m with (Line 175) “slope of line=3.49”

L 120 and L128-129 and L 133-134 seems to be redundant. Consider reformulate or explain why these two statements are not redundant

→ Good catch, removing the second statement.

L 137-157. This could maybe be shortened and included later in the text? E.g. along with the corresponding results. Those are controls to show your results are robust to different technical factors, but we have not read yet the results. Maybe consider reorganizing (this is only my personal opinion).

→ Agree that the order might be a little obscure. We debated the placement of this actually and we understand there might not be a ‘right’ answer. If it goes later on with the beta diversity section, we could see how some readers would be confused and then question if this metric was also applied to alpha diversity. Since it is part of the initial post-processing QC and ultimately justification of how the samples are handled for all analyses going forward, we decided to place it at its current spot. We’ve added more details to deescalate unwanted attention towards the finer details of the actual beta diversity results.

(Line 203-204)

“The details of the actual beta diversity statistical significance of the various fish related metadata is discussed later. “

*Alpha

L160. Seems like you also include results for microbial biomass estimation. This could be acknowledged in the title of this section, for clarity.

→ Great point, we’ve changed to clarify (Line 206)

“Factors influencing alpha diversity and biomass of the fish microbiota”

*Beta diversity and phylosymbiosis

L 240. Typo: “Body sitel”

→ Fixed, (Line 286)

“metrics for individual body sites. “

L 241-243. This is a very interesting test (Fig. 5b), but it was not really put in context in the introduction.

→ Based on the feedback of this review, we've restructured our intro and have added specific details about the trophic level hypothesis. In short, one idea is that herbivorous animals have evolved much longer guts to aid in digestion and thus one could hypothesize that a longer gut would mean a greater difference of microbial communities between the start and the finish especially if the physiology changes due to hindgut fermentation.

(Line 106-115)

“Environmental factors such as seasonality, salinity, and geography along with diet can influence the microbiota compositions in fish ^{10,11,25}. Gut microbiota of vertebrates including fishes are often differentiated by trophic level for beta diversity ^{11,26–28}. Gut lengths are longer in herbivorous fishes which is hypothesized to increase digestion efficiency but also differ across habitats ^{29,30}. Hindgut fermentation in the guts of herbivorous fishes influences the microbial community composition ³¹. In soil-associated invertebrates, higher trophic level was linked with higher microbial diversity and novel microbial species ³². In humans however, increased fecal microbial diversity is associated with higher consumption of plant diversity ⁹. One outstanding question is if microbial diversity is higher in lower trophic fishes?”

L 250-L255. You might be able to reduce the length here.

→ We've trimmed this down

L261: Prediction → “predictive ability”

→ Changed (line 304)

“abundances had a stronger predictive ability of host genetic similarity”

L 267. Do you have an empirical reference for co-evolution?

→ Change to (line 312-316)

“ If vertebrates have co-evolved to some extent with their microbiota, one could speculate that strains from genera that contain known probiotics found in fishes would have higher performance (improved ability to adhere and colonize to mucosal environments) in fishes as compared to the application of allochthonous terrestrial-derived probiotics applied to fishes.”

L 268-269. What do you mean by “higher performance” ?

→ We've changed this sentence to clarify. One of the big challenges with probiotics is ensuring that the bacteria actually sticks around and thrives within the host rather than just passing through. This may partially have to do with a bacteria's ability to bind to and colonize the mucosal environment. There are more examples in the pathogen literature but Justin Sonnenberg has published extensively on how microbes colonize the gut in mammals and the importance of glycobiology in explaining some of these interactions.

PMID: 34425698, PMID: 15790854, PMID: 15164016

(line 312-316)

“ If vertebrates have co-evolved to some extent with their microbiota, one could speculate that strains from genera that contain known probiotics found in fishes would have higher performance (improved ability to adhere and colonize to mucosal environments) in fishes as compared to the application of allochthonous terrestrial-derived probiotics applied to fishes.”

L 267-278. While interesting, this section on probiotics is rather enigmatic to me. It is not really introduced or justified anywhere. I would consider removing it to keep a clearer focus.

→ We've added two sentences in the beginning of the paragraph to hopefully better bridge the concepts. We understand from a basic research perspective, it may not be of interest. Since NCOMMS is broadly read, probiotic ecology is something of interest to many researchers looking for direct applications to their system of interest. From a translational perspective (e.g. application to aquaculture), we believe this is a critical observation and gives a foundation for future research to actually test these hypotheses on these strains across vertebrate classes or even within fish species.

(Line 310-316)

“Since our data show that fishes that are more similar to each other (shorter phylogenetic distance or branch length) have a more similar skin, gill, and hindgut microbiota, we wanted to explore how phyllosymbiosis could be used in the discovery of probiotics. If vertebrates have co-evolved to some extent with their microbiota, one could speculate that strains from genera that contain known probiotics found in fishes would have higher performance (improved ability to adhere and colonize to mucosal

environments) in fishes as compared to the application of allochthonous terrestrial-derived probiotics applied to fishes.”

*Source pools

L313-315. Very interesting hypothesis. Can you more explicitly say if you refute it or not?

→ Thanks for the kind remark. We’ve added additional sentences at the end of this paragraph to summarize that finding and give an alternative hypothesis for follow up work.

(Line 366-388)

“Based on this observation, ‘flow refuging’ fish which includes flatfish indeed have a higher proportion of their skin microbes originating from the sediment whereas the fast swimming fishes (pelagics and coastal pelagics) have most of their skin microbes originating from the water column. One aspect we were not able to compare was the influence of prey items. Based on diet surveys, marine invertebrates were consumed by 107 species of fish while fish were the second most common prey item (Supplemental Figure 5a). Within the invertebrates consumed, arthropods, mollusks, and annelids were the most consistent prey items (Supplemental Figure 5b) all of which have many species globally (Supplemental Figure 5c). By comparing the ratios of microbial origin environments (e.g. sea water vs sediment), we can discover new insights into host-associated microbial ecology in the ocean. Microbial origins of marine fish mucosal remain largely unknown, but ecological factors, such as habitat type, may influence the extent of microbial colonization from sea water or sediment.”

*Gamma diversity

L 325-326. Can you explicit a bit more your hypothesis here? Why is it that an “older evolutionary history” implies more gamma diversity?

→ We have added two clarifying sentences to help guide the reader through our logic

(Line 386-389)

“ If animals have co-evolved with their microbiota, one could hypothesize that older lineages will have had more time to optimize these microbial relationships. If microbiota colonization is random, then one would expect diversity to simply be a function of the environmental exposure.”

L 364. I am not sure to understand which model you are referring to here. How this model was constructed? The associated figure shows a Michaelis-Menten model, but I don't find it in the methods. This needs a better explanation.

→ We've added additional details for the methods. It was a mistake to not have added that in the previous version.

Briefly, we modeled the fit of the line with a Michaelis-Menten model which assumes saturation at some point in the future. Its possible there are better models and its also possible we need a lot more data to have a better prediction. Despite these caveats, we wanted to model our data to have some ballpark estimate of how much more sampling is needed. After getting 1000 fish species in a future study, someone may update this number only to learn that we actually need 20,000 species to reach saturation etc.

(Line 886-893)

“To estimate the total number of unique ASVs (sum of unique ASVs across the four major body sites within a fish) which exist across all fish species, we fit a Michaelis-Menten equation to the results from Figure 8f using Prism v9. This equation will determine the number of fish species at which approximately half of the microbial diversity will be found (Km value). In addition, the model will produce the maximum y value (Vmax value) which in this example would be the total number or maximum number of unique microbial ASVs. This extrapolation is an estimate which should be updated as more data is added through future experiments or meta-analyses.”

L 363-364. I don't know what an “old” fish could be, and I don't understand why you ordered the species like that.

→ Clarified (Line 425-427)

“The fish species were arranged on the x-axis by class (with earliest lineages first: Myxini, Chondrichthyes, Osteichthyes) and then alphabetically by order and family.”

L 375-376. Can you explain how microbial biomass estimations feeds into your estimate of fish gamma diversity?

→ Thanks for making this point. We've updated the section to clarify how using biomass estimates will help future researchers better estimate absolute microbial diversity with gamma diversity being just one example. (Line 436-441)

“For future research on gamma diversity across other vertebrates, we recommend comparisons of multiple body sites along with the integration of microbial biomass estimates as demonstrated in this study. The integration of microbial biomass enables a more accurate assessment of total absolute microbial diversity in a sample. By using our method, one uses all the reads in the sample to estimate diversity rather than needing to rarefy which forces one to lose or disregard sequences leading to underestimates of true diversity.”

**Discussion

L 401-402. Can you speculate / give hypothesis why this could be the case?

→ Thanks for this feedback. Despite the length of the ms, we've added additional details/hypotheses here. We think this is an interesting area of new research direction and try to highlight / drive home the point.

(Line 490-510)

“Microbial biomass, habitat depth 1 (shallow ‘neritic’, mid water ‘mesopelagic’, or deep-sea ‘bathypelagic or abyssal’), and substrata group (pelagic, soft bottom, rock associated, deep water benthic) were also top predictors of beta diversity across the entire dataset suggesting that habitat had an influence on the microbiota of fishes. Microbiota diversity, function, and stability is known to vary across water column depth as a response to decreasing sunlight, availability of organic matter and nutrients mediated by the microbial loop, mixing, oxygen concentration, and water temperature^{42,43} Thus, deep-sea fishes, especially those which have limited vertical migration, would have different microbial exposures at depth. The influence of substrata group on fish microbiota diversity was an interesting observation. We speculate that the benthic structure likely influences the sediment microbiota and that in turn influences resuspension of microbes and remineralization of nutrients back into the water column. Benthic complexity is a key driver of local fish diversity (e.g. kelp forest vs inner slope) thus animal assemblages themselves will differ across varying substrates⁴⁴. Conversely, the diversity of fish or animal assemblages will influence the diversity and magnitude of organic wastes which get deposited to the benthos thereby impacting local sediment microbial diversity. Fishes, particularly flatfish, can also influence the benthic sediment by physical resuspension⁴⁵. To date, sediment microbiology research has focused on impacts to free-living communities in the overlying water column, with less attention to impacts on the overlying animal assemblages and vice versa. We see this as a promising area of future research to better understand the microbial loop and sediment contributions to host-associated microbial biodiversity.”

L407. is microbial fouling a known problem to maintain respiratory performance? Can you provide any reference?

→ This is still speculative which is why we used the word hypothesis. The basis of the hypothesis however is derived from parasite ecology in Tuna farming/ranching. Parasite egg count abundance in the gill is associated with pathology in tuna production at

various life stages. We've added some references though for this.

(Line 530-534)

“keeping gills clear of microbial fouling to maintain respiratory performance. Previous research on *Thunnus maccoyii*, Southern Bluefin Tuna, has shown that parasitic infection with blood flukes, *Cardicola* spp., results in pathogenesis of fish from both fluke and egg infections of the gill⁵². Elevated densities of parasite eggs in the gills leads to gill tissue damage, hypoxia and mortality⁵³.

L 414-415. Can you tell us more how this impact the interpretation of your results?

→ added a sentence and reference (Line 519-523)

“Differences between the midgut and hindgut emphasize the need to further describe the physiological or abiotic factors (e.g. oxygen) micro gradients within the GI tract of fishes⁴⁹. The gastrointestinal tract can vary in morphology and function across fishes with many differing types of stomachs, lengths, and enzyme profiles^{30,50,51}. “

L 433-438. These seem more related to the methods. If you need to save space, this could be shortened.

→ Thanks, moved to methods (Line 735-748)

“We estimated trophic level by two primary methods. First, we estimated the trophic level by using previously documented diet data derived from the literature. Diet preferences of each species was determined through literature searches including secondary literature databases such as fishbase.org. Where diets differed for juvenile and adult stages, all components were included in the metadata. A coarse estimate of trophic level was determined based on the general diet. If a fish rarely eats fish or if fish are a minor component of the diet, they are still considered in the trophic numbering where 1=herbivore (consumes primary producers), 2= primary carnivore (consumes primary consumers such as zooplankton or other invertebrates), 3= secondary carnivore (consumes small fishes), 4= tertiary carnivore (consumes large carnivorous fishes), or 5= quaternary carnivore (consumes high trophic fish or marine mammals). These values can be found in the metadata column (trophic_likely). For the second method, we used the ratio of 'relative intestinal length : total body length <RIL:TL>' as an indicator of trophic level⁵⁵. Fishes that are more herbivorous will have a higher RIL greater than 1 and upwards of 5-30 whereas carnivorous fishes will generally be much lower < 1²⁹.”

L 418. Do you mean an evolutionary role of the gill microbiome for its host? Please clarify.

→ Agree that was a little confusing - we have modified

L 471-474. Very interesting idea: can you elaborate a bit more on it?

→ Thanks, we have clarified this statement and given definitions (Line 587-591)

“For gut comparisons, future studies should aim to investigate the importance of reproductive strategies in fishes (viviparity 'internal fertilization and live birth' vs ovoviviparity 'internal

fertilization' vs. oviparity 'external fertilization') to determine if phylosymbiosis and potentially co-phylogeny is stronger in fishes which utilize viviparity.”

L 473. By “co evolution”, do you mean “co phylogenies”?

→ We understand its a fiercely debated topic and do not wish to misrepresent our data, thus we have changed our wording. (Line 590-591)

“and potentially co-phylogeny is stronger in fishes which utilize viviparity.”

L 474-476. Can you explain why this might be important?

→ Added a sentence on this (Line 593-596)

“With a large and recent radiation, it may be possible to further tease apart and test these hypotheses regarding the mechanisms around stochastic microbial exposure leading to potential adaptations in the host leading to co-phylogeny.”

L 482-483. Can it be that those microbes are endemic to the fish species you are studying, so there are no “external” sources to look for?

→ Indeed, you are absolutely correct. We’ve highlighted this now with an extra sentence.

(Line 611-613)

“ It is also possible that these microbes are simply endemic to the body sites of the fishes from this study, which highlights the importance of integrating concepts of microbial ecology into conservation biology.”

L 491. What do you mean by “diverse evolutionary history”? Could you be more specific?

→ Changed “diverse” to deep (Line 621-622)

“deep evolutionary history”

*Figures

L1039. Evolutionary divergence is a DISsimilarity (not a similarity). Please clarify.

→ Changed, (Line 1333-1334)

“Effect of evolutionary distance (low divergence time indicates a short branch length or similar fish species) of all fish compared

Fig 6 c and d seems to have the same Y axis label. Please correct

→ Good catch, thanks for pointing that out. We have corrected this (D)=sediment

Reviewer #4 (Remarks to the Author):

This study presents an impressively large data set with a diverse composition of marine fish species. However, while the work behind generating this data set and the high number of species included is impressive, I miss a more concrete and underlying hypothesis to be tested. Hence, this manuscript comes across as a very descriptive study that uses an impressively large data set to draw a range of 'random' correlations based on available data rather than more specific hypothesis driven analyses. This way, the discussion also often seems to just repeat the results section rather than highlighting the broader relevance of truly novel findings.

→ We have spent considerable efforts reformatting the intro to better frame our hypotheses. We have also changed and reformatted the results and discussion. However, we also would like to emphasize that a study of this magnitude is indeed foundational to the literature and as a result will generate many hypotheses from this. We do not see this as a fundamental problem in the research goals and believe that many will use this dataset in the future to explore and test some of the hypotheses put forth as we are in subsequent upcoming manuscripts.

For example, that species, body site and diet all modify microbiomes is not surprising given the existing literature; so still little is revealed as to what are the functional and evolutionary consequences of these observations among body sites etc.?

→ The majority of the existing literature focuses on analyses of one fish at a time. We've cited most of the larger experiments which consider multiple fishes and multiple body sites. Even fewer studies will look at a combination of multiple species of fish along with multiple body sites, therefore cementing this dataset as truly novel and foundational to the field. We address the point about functional microbiomes below. As for the evolutionary consequences, we agree this is an important point and discuss this in the

discussion. Our dataset puts forward several novel hypotheses for potential evolutionary trade-offs with the host-microbiome which can be tested in the future.

Another example is Figure 3 which is kind of underwhelming and not very surprising as some correlations would always be expected with so many samples. That diversity varies among body sites, species and environmental conditions is not very surprising either; contrary what is the relative effect of body site compared to other explanatory variables?

→ When the reviewer says: "some correlations would always be expected with so many samples", and then discusses body sites, we are a little confused as to what they are actually trying to say. The only correlations performed are in Figure 3f which is looking at various biometric measures, swim performance measures, and microbial biomass against alpha diversity and microbial biomass. In 3f, you can compare the relative effects of these measures by comparing the rho values across body sites or within a body site. With correlations, it's better to have larger sample sizes. Our sample sizes are really not that large. If we had 10,000- 100,000 samples then one would be more concerned with spurious significance (type 1 error). Since we saw that body sites differed in microbial diversity, we analyzed body sites independently for the proceeding measures. This is a common practice in microbiome research to reduce noise and essentially ask if these environmental or biological effects impact body site microbiota equally. Some of these findings may not be surprising to the reviewer, but most of them have never been tested or at least published.

All of this data is present in the existing figures. You can get a sense of the relative effect by looking at the H statistic for instance when there is significance. You can quickly tell that the body site has the greatest effect for each of the alpha diversity metrics but is not significant for microbial biomass (Figure 3d).

In addition, crucial information relating to the data generation is completely lacking including what microbial DNA markers were amplified and how ASVs were defined (see comments below).

→ We've updated this (see below): 16S rRNA V4 region 515/806 bp → Deblur (exact sequence variants)

Overall, my feeling is that despite the large data set in terms of number of species, the DNA analyses lack critical information and little new information is revealed in terms of fx the functional and evolutionary role of fish microbiomes. My concern is also illustrated by the descriptive nature of the three research questions put forward in the introduction, which I comment here (author's text listed in quotes):

→ We've updated the information on how samples were processed for 16S sequencing. Our study focused on the microbiota. We agree that looking at the metagenomes to assess the functional role would be great, but currently there are grand technical challenges in doing so. The primary challenge in doing host-associated metagenomes is the problem of host contamination. Although several studies have sought to overcome this, most of these methods only work for fresh liquid samples such as saliva [PMID: 29482639]. Attempts on fish microbiomes have been tested without success (unpublished). As for the descriptive nature of the research questions, we have made substantial changes in the form of the intro and our hypotheses to better frame the study.

Lines 86 – forward:

“This study aimed to answer three main questions to understand the ecological and biological drivers of the fish microbiome:

1) what are the primary factors that influence host-associated microbial communities for marine fishes (e.g. body site location, habitat, trophic level, swimming method, phylogeny, etc.)”

Here, you only draw correlations from available data (i.e. not hypothesis driven) while not including e.g. host genetics in more detail than the phylosymbiosis tests. This way I do not feel that the study is very conclusive as to the bigger picture of this question. What would be more novel is to sequence some host genomes to dig deeper into understanding what host genes are driving the phylosymbiosis patterns to better understand how the host constrains microbiomes in relation to environmental variance.

→ This is the first paper to test phylosymbiosis across multiple body sites in a controlled manner. We make an important observation that both the gill, skin, and hindgut communities exhibit some sort of phylosymbiosis signal. Indeed the next step would be doing comparative genomics of the fishes to try and understand this in context to adaptations and evolution. Sequencing 116 fish genomes would be fantastic but it is very expensive and would be outside the scope of this study. It isn't worth while to sequence Eukaryote genomes unless you are going to meet the requirements of a high quality genome put forth by the VGP [PMID: 33911273]. Sequencing a few fish genomes really won't be that informative because you won't have the power to make any conclusive statements. The study proposed to look at how the host constrains the microbiomes in relation to environmental variance would be a great study and we certainly believe that that should be a valid way to test some of the hypotheses generated in this study. We did add some sentences on this in the conclusion:

(Line 695-700)

“Finally, we make the case for the fish genome community to come together with the fish microbiota community to unify efforts, resources, and data sharing to accelerate discoveries. Testing hypotheses around host-microbiota interactions will only be feasible with more comparative genomics studies and high quality reference genomes of fishes to complement our evolving appreciation of the fish microbiota.”

“2) where might these microbes originate (sea water, sediment, host, etc)”

This is an interesting question, but the analysis and results addressing this seem inconclusive when they merge so many different species, as it is not very surprising that the surrounding environment always contributes some microbes, but maybe it is not the important ones? Also, it is not clear if some species in the seawater were derived from fishes and not vice versa.

→ We have expanded this section in the discussion to speculate the origins but also expand on the result that many of the microbes on the fish may indeed be endemic to the fish host and rare in the free-living environment.

“3) is microbial diversity greater in fishes compared to other vertebrates considering fishes are evolutionarily more ancient.”

→ We have changed the sentence structure to reflect the experiment

In the intro we've changed it to "is microbial diversity greater in fishes compared to other vertebrates considering fishes are more genetically diverse"

In the results we've changed to

" If animals have co-evolved with their microbiota, one could hypothesize that older lineages will have had more time to optimize these microbial relationships. If microbiome colonization is random, then one would expect diversity to simply be a function of the environmental exposure."

The rationale of this question is also not clear – why would more ancient species have higher microbiome diversity? Couldn't it be the opposite that ancient species have had time to evolve their own genome in a way that more effectively selects for only relevant symbiotic species at the expense of random microbes from the environment?

→ We have clarified this more, but honestly it is an open ended question. The most basic premise originates from the observation of unique microbiomes associated with individual species. One can expand this and hypothesize that as you increase in species you will also increase in microbial diversity. We've noted this alternative hypothesis in the discussion.

(Line 620-624)

"Our study showed that amongst vertebrates, fishes have the most unique assemblage of total microbial diversity, which we hypothesized corresponds to the deep evolutionary history and habitat types occupied by fishes. An alternative hypothesis is that more evolutionary ancient lineages of animals would have had more time (generations) to optimize and reduce their microbiome complexity."

Thus, I think a more specialised journal would be a better fit for this large and useful data set in its current form as it mainly presents value to the fish community. Alternatively, I advice the authors to analyse the data in a more concrete hypothesis driven manner that e.g. addresses more general evolutionary aspects such as the relative roles of environment vs host genetics in shaping mucosal microbiomes.

→ We thank the reviewer for their comments. We have taken the approach to reformulate and reframe our experiment into a more concrete hypothesis driven

approach which is now setup more clearly in the intro and throughout the manuscript. We believe this manuscript has a great fit with the broad audience of NCOMMS. The value of this article is not just for fish researchers, although there are plenty of those, but also for general host-microbiome researchers. Our method surrounding the biomass estimation is applicable to all microbiome studies and believe this is a great demonstration of that tool. We have made substantial changes and hope the reviewer will be pleased with these edits as recommended.

Other comments:

Throughout: change “isn’t” to “is not” and for similar abbreviations

→ Changed, we only found 1 place in the manuscript with isn’t

<>

The method to estimate biomass is interesting, but it is not an essential part of the story being told. Further, there is no information given as to if (and how) the method considers copy number variation of 16S genes among bacterial species assuming the method is based on 16S amplicon sequencing (see next comment).

→ The biomass estimation is a critical component of the story and we have highlighted this in various areas including the discussion. CNV is an important point but it turns out its really hard to do and most studies which attempt to perform a normalization, end up adding more noise. There is a great benchmarking paper on this which we’ve also cited in the discussion PMID: 29482646. Therefore, it would be inappropriate to attempt to normalize for CNV at this time. Microbial biomass is an important and often neglected measurement in microbial ecology studies because it is so difficult to measure. This study now enables anyone to implement this method into their microbiome experiment. This method estimates the microbial biomass by estimating the copies of 16S per gram of animal tissue or mucus. Here we describe the use of this method in a new paragraph of the discussion.

(Line 456-481)

“One of the biggest challenges in microbial ecology is estimating microbial biomass also known as total microbial load or absolute abundance. This is especially challenging in host-associated environments as the existing methods to do this including microscopy, flow cytometry, and qPCR³⁴ are optimized to work best on free-living systems and may have lower throughput or require immediate processing. Flow cytometry methods are challenging due to excessive clumping of cells, lower throughput, and require immediate processing of samples³⁴ while qPCR assays (e.g. 16S) can have many false positives due to contamination and thus co-amplification of either chloroplasts or mitochondria³⁵. Another approach is to use a spike-in of known cell counts of control bacteria³⁶ or synthetic oligos³⁷ into your samples which can enable biomass estimations post sequencing. The downside to this approach is you lose sequencing depth across samples and if using cells, it is necessary to ensure that those microbes are not already found in your samples. Many studies ignore estimating biomass and instead are forced to assume equal biomass across samples. This strategy causes problems when trying to compare relative abundances of microbes across samples which has led to many computational tools being developed around compositionality^{38,39}. Our method is extremely useful and novel in that it enables one to estimate the microbial biomass (namely number of 16S copies per gram of tissue) using standard 16S high throughput sequencing and without contaminating samples with a spike-in. One caveat is that many microbes have multiple 16S copies per genome, thus one may want to perform this normalization on their sample set in the future if *rrn* copy number is known across the resident microbial taxa. Attempting to normalize per 16S copy number per genome however is challenging and most studies actually fail to do this, thus we have avoided this additional normalization attempt in our workflow⁴⁰. Our results directly refute the hypothesis that sequencing depth is random and instead show that it is highly correlated to the microbial biomass. We leverage this observation and provide a framework for consistent sample processing to enable insights into microbial biomass which should be applicable across all ecosystems including animals, plants, soil, water, and the air.

L.643-648: In this part of the methods, you should also specify what DNA has been amplified in the PCR? I assume it is some type of 16S based primers, but there is no information given regarding what is amplified, and this is one of the most important details for the entire protocol.

→ We've added this information. Previously we had just cited the method and indicated we used the earthmicrobiome project protocols but we realize not everyone might be familiar. (Line 800-802)

“For the PCR step, a total of 5 ul of DNA was amplified in a miniaturized 10 ul PCR reaction using the standard EMP protocols for the V4 region of the 16S rRNA gene 515F/806 Rb^{85,86}.”

L.679: There is also no description on how ASVs were defined, and which similarity thresholds were used?

→ We describe this in an earlier section: In short we used deblur and this is one of the ~2 exact sequence variant methods out there.

(Line 811-813)

“Samples were uploaded and processed computationally using Qiita⁸², trimmed to 150 bp and ASVs generated using Deblur v1.1.0 and only ASVs passing the positive filter step were used⁸⁷.”

L.733: You may want to temper this title to discovery of probiotic candidates. Also, the first argument in this paragraph is that probiotics of terrestrial origin are not good representatives for fishes. Then you still decide to only look for groups that are well known probiotics in terrestrial animals; I would question that this is the best way to discover potentially new useful probiotics in fishes?

→ We agree about the title, we changed to (Line 904)

“Discovery of probiotic candidates”

→ As for the focus on Bacillus and Lactobacillus, the point is that there are many strains within these genera which are essentially broad spectrum probiotics. Many of these have even been tested in aquaculture experiments. Are there better ones for fish? Most likely, but we don't necessarily know what those are yet. One of the benefits of these large datasets, is that as researchers find a new candidate probiotic, they can then go and mine the datasets to see which fish species or body sites it might be found in.

We've added a few sentences at the end of the phyllosymbiosis discussion (within discussion) to address this point.

(Line 596-603)

“In light of the observation of possible phyllosymbiosis occurring across multiple body sites, we encourage probiotic researchers to consider this when looking for new candidate strains for aquaculture. For example, we show that both Bacillus and Lactobacillus are present across multiple species and body sites, but it is possible that other, unexplored microbial taxa may be effective candidates. Datasets like this can be used to mine and identify where new strains might be residing on the hosts, although future work is necessary to define the functional

contributions of fish-associated microbiota using metagenomic and metatranscriptomics methods in addition to cultivation campaigns.”

RE ST2 analysis... where are the fish microbes coming from?

→First and foremost, they could simply be unique to the fish and not found in the free living environment, at least not at high levels. To consider all possible sources of microbes in the marine environment (aside from sea water and sediment), we went back to our list of fishes and determined the prey types. We used fishbase and other supporting databases or primary literature to get the estimates. We sought to evaluate the diversity of prey items and further compare this to the actual known diversity of these prey items in the global oceans. The most common prey item was invertebrates which is consumed by 107 out of the 118 fishes (Supp Fig 5a). The second most common prey item was fish with algae and macroalgae along with marine mammals the least common. Within the marine invertebrates, arthropods, mollusks, annelids, and echinoderms are all commonly consumed (Supp Fig 5b). The marine biodiversity estimates of all prey items demonstrate the importance of microbiome surveys from these communities.

We have added details to the results

(Line 370-376)

“Based on diet surveys, marine invertebrates were consumed by 107 species of fish while fish were the second most common prey item (Supplemental Figure 5a). Within the invertebrates consumed, arthropods, mollusks, and annelids were the most consistent prey items (Supplemental Figure 5b) all of which have many species globally (Supplemental Figure 5c). By comparing the ratios of microbial origin environments (e.g. sea water vs sediment), we can discover new insights into host-associated microbial ecology in the ocean.”

Discussion (Line 611-617)

“It is also possible that these microbes are simply endemic to the body sites of the fishes from this study, which highlights the importance of integrating concepts of microbial ecology into conservation biology. As animal species go extinct, it is possible their microbiota will also follow. Prey items also contribute to the microbial diversity in fishes⁶⁹. A large-scale marine microbial sampling effort should include diverse sediment types, seawater from bays and offshore environments, as well as representatives from the thousands of marine invertebrates, including plankton, and macroalgae species.”

Supplemental Figure 5. General prey diversity across 118 fish (116 species) samples. Prey types of each fish determined using fishbase.org or primary literature. a) Total number of fishes (of the 118) which consume phytoplankton or algae, invertebrates, fish, or mammals. Prey data for 3 species of fish were not available (unknown). b) Higher resolution breakdown of prey consumption of fishes. c) Biodiversity estimates from the various algal, invertebrate phylums, marine fish, and marine mammal groups.

Reviewer #3 (Remarks to the Author):

I am reviewer #3 from the previous round of review.

General comment on the revised MS

In this revised version of the manuscript, the authors substantially reframed the introduction to better put in context the knowledge gap and their associated research question and approach. I think this revised introduction reads better than the previous one. Please find below additional comments that you might want to consider in order to improve the clarity of the MS.

Specific comments on the revised MS

L 87 – Phyllosymbiosis is the association of microbial *composition* with host phylogeny

L 89-92. This is a very interesting information, but it lacks connection with the previous statement about phyllosymbiosis. Do you suggest that because fish are more exposed to environmental bacteria, they are expected to have a different level of phyllosymbiosis?

L 115 – “In fishes from lower trophic levels” ?

L 125. Typos (2 commas)

L 310-325 This section on probiotics (and other mention of it) remains enigmatic to me, especially when it is linked to phyllosymbiosis.

L 387. Here, I am missing how “optimization” of microbial relationship relates to gamma diversity.

L 579. Not sure the use of “obscure” is necessary here

L 613. I am not exactly sure how “conservation biology” relates to the previous sentence.

L 667. A point is lacking between the two sentences.

All Figure (style) – For ease of reading, maybe worth removing underscores from title axes

One of my previous technical comment was probably unclear so I repeat it here and try to better explain the point:

(refers to the method section on the correlation between alpha diversity and host traits) “Usually, when using species data in correlative analysis, one should check for the presence of phylogenetic signal in the residual of the models (to avoid phylogenetic pseudo-replication). This can be for example estimated with Pagel Lambda. Useful references include Felsenstein, J. Inferring Phylogenies. (2004); Uyeda, J. C., Zenil-Ferguson, R. & Pennell, M. W. Rethinking phylogenetic comparative methods. Syst. Biol. 67, 1091–1109 (2018); Pagel, M. Inferring the historical patterns of biological evolution. Nature 401, 877–884 (1999); Revell, L. J. phytools: an R package for phylogenetic comparative biology (and other things). Methods Ecol. Evol. 3, 217–223 (2012). »

What I meant here relates to the *host* phylogeny, not the microbial one. When you carry correlation between two host features (here microbial alpha diversity and some host trait), you should in theory check that there is no phylogenetic signal in the residuals of the model (check the references given above). This is also known as phylogenetic non independence and that is why some of the phylogenetic comparative methods have been developed.

Reviewer #4 (Remarks to the Author):

The authors have done a good, and impressively large, effort in revising this manuscript. They have addressed all reviewer comments and especially the introduction and discussion now appear stronger and more interesting in light of the presented hypotheses.

While the authors have addressed most comments satisfactorily - often by adding whole new paragraphs - this version of the manuscript (still) appears overwhelmingly long and wanting to cover many different things so that one gets easily lost regarding what are the main messages. I reckon this may simply be an unavoidable outcome when presenting this large data set as a single all inclusive publication.

Minor comments:

L. 579: Maybe use different terminology than "obscure" when talking about other studies' methods.

REVIEWERS' COMMENTS

Reviewer #3 (Remarks to the Author):

I am reviewer #3 from the previous round of review.

General comment on the revised MS

In this revised version of the manuscript, the authors substantially reframed the introduction to better put in context the knowledge gap and their associated research question and approach. I think this revised introduction reads better than the previous one. Please find below additional comments that you might want to consider in order to improve the clarity of the MS.

Specific comments on the revised MS

L 87 – Phylosymbiosis is the association of microbial *composition* with host phylogeny
→ Line 87: Changed to “, the association of microbial composition with the host phylogeny”

L 89-92. This is a very interesting information, but it lacks connection with the previous statement about phylosymbiosis. Do you suggest that because fish are more exposed to environmental bacteria, they are expected to have a different level of phylosymbiosis?

→ Exactly, we thank the reviewer for pointing out our lack of clarity here. We've added a final sentence to connect the two concepts to drive home the point

Line 92-95

“Thus, since fishes are exposed to much higher concentrations of microbes throughout their even longer evolutionary history, it is likely their level of phylosymbiosis will differ from terrestrial vertebrates.”

L 115 – “In fishes from lower trophic levels” ?

→Line 118, Changed, “One outstanding question is if microbial diversity is higher in fishes from lower trophic levels”

L 125. Typos (2 commas)

→ Line 131: fixed

L 310-325 This section on probiotics (and other mention of it) remains enigmatic to me, especially when it is linked to phylosymbiosis.

→ NCOMMS will have readers who are both from a basic research background that don't care at all about applications such as probiotics, but there are also readers from applied animal science backgrounds. The finding of evidence for phylosymbiosis may

be important for animal scientists who are trying to figure out better ways of developing probiotics. If you are a fish farmer and use probiotics in your feeds, it is likely that those strains are derived from cattle or some terrestrial system. Those strains may not be able to adhere to or successfully colonize the fish gut as easily as a strain that is derived from a fish. Overall, this is a novel concept and will need to be followed up on with specific experiments to test the impact. Another application would be FMT researchers treating patients with C diff. Yes, folks can use human feces (donor) to treat C diff in mice but a mouse (donor) will likely have an improved effect if the recipient is a mouse. We choose 2 of the most commonly used probiotics (Bacillus and Lactobacillus) to determine if they are even present in fish. It is entirely possible that other strains from completely different orders or Phyla are more important. Hopefully this clarifies the point.

L 387. Here, I am missing how “optimization” of microbial relationship relates to gamma diversity.

→ Line 395-396: This was in response to another reviewer. I've added a line “This could include removal of detrimental taxa or enrichment of beneficial taxa.”

L 579. Not sure the use of “obscure” is necessary here

→ removed ‘obscure’ although for the record – the manuscript does not give any rationale as to why they did that or at least I did not understand the down sampling methods

L 613. I am not exactly sure how “conservation biology” relates to the previous sentence.

→ Line 623-624: added a sentence

“That is, if a fish goes extinct, so may the microbiome associated with that species of fish.”

L 667. A point is lacking between the two sentences.

→ Line 678, thanks! Added

All Figure (style) – For ease of reading, maybe worth removing underscores from title axes

One of my previous technical comment was probably unclear so I repeat it here and try to better explain the point:

(refers to the method section on the correlation between alpha diversity and host traits)

“Usually, when using species data in correlative analysis, one should check for the presence of phylogenetic signal in the residual of the models (to avoid phylogenetic pseudo-replication). This can be for example estimated with Pagel Lambda. Useful references include Felsenstein, J. Inferring Phylogenies. (2004); Uyeda, J. C., Zenil-Ferguson, R. & Pennell, M. W. Rethinking phylogenetic comparative methods. Syst. Biol. 67, 1091–1109 (2018); Pagel, M. Inferring the historical patterns of biological

evolution. *Nature* 401, 877–884 (1999); Revell, L. J. phytools: an R package for phylogenetic comparative biology (and other things). *Methods Ecol. Evol.* 3, 217–223 (2012). »

What I meant here relates to the *host* phylogeny, not the microbial one. When you carry correlation between two host features (here microbial alpha diversity and some host trait), you should in theory check that there is no phylogenetic signal in the residuals of the model (check the references given above). This is also known as phylogenetic non independence and that is why some of the phylogenetic comparative methods have been developed.

→ We acknowledge that this is an important caveat. We have added two sentences in the discussion on this.

Line 574-579

“Future work on associations of alpha diversity and environmental signals such as habitat or fish physiology like swim performance should strive to include a larger replication of species across distant phylogenetic groups to avoid potential biases of phylogenetic pseudo-replication. For instance, although we observe decreasing microbial biomass in the gills of pelagic fishes, it is possible this signal is driven by some evolutionary adaptation which was developed in Scombrids but not in other pelagics.”

While this does seem useful, we have not seen this sort of analysis done in microbiome analyses across broad phylogenetic groups of animals so are hesitant to introduce it now. It is possible we could have some phylogenetic signal for certain groups (e.g. the Scombrids are all pelagics/coastal pelagics and are in the fast swimming groups etc.). While 101 species is a lot, we absolutely agree that having larger sample sizes in the 1000s will be required to dive into and test many of these hypotheses generated. Ultimately, we agree that evolution of the host-microbiome interface is an exciting area of research which can take advantage of this dataset but we do not believe it is appropriate to apply here. It could be an entire new manuscript all together.

Reviewer #4 (Remarks to the Author):

The authors have done a good, and impressively large, effort in revising this manuscript. They have addressed all reviewer comments and especially the introduction and discussion now appear stronger and more interesting in light of the presented hypotheses.

→ We are grateful to have such thorough and helpful reviewers and thank reviewer 4 for their help in making this a better manuscript.

While the authors have addressed most comments satisfactorily - often by adding whole new paragraphs - this version of the manuscript (still) appears overwhelmingly long and wanting to cover many different things so that one gets easily lost regarding what are the main messages. I reckon this may simply be an unavoidable outcome when presenting this large data set as a single all inclusive publication.

→ In addressing all of the reviewer's responses, the manuscript has indeed increased in length. We don't really see many ways of reducing it without substantially missing some of the criticisms of the reviewers.

Minor comments:

L. 579: Maybe use different terminology than "obscure" when talking about other studies' methods.

→ As noted by the other reviewer as well, we have changed this.